



# Subsurface CO₂ dynamics in a temperate karst system reveal complex seasonal and spatial variations

Sarah Rowan[1,2], Marc Luetscher[3], Thomas Laemmel[1,2], Anna Harrison[4], Sönke Szidat[1,2], Franziska A Lechleitner[1,2]

[1] Department of Chemistry, Biochemistry, and Pharmaceutical Sciences, University of Bern, 3012, Bern, Switzerland.
[2] Oeschger Centre for Climate Change Research (OCCR), University of Bern, 3012, Bern, Switzerland
[3] Swiss Institute of Speleology and Karst Studies, 2300, La Chaux-de-Fonds, Switzerland.
[4] Institute of Geological Sciences, University of Bern, 3012, Bern, Switzerland.

*Correspondence to*: Sarah Rowan (sarahrowan96gmail.com)

**Abstract.** Understanding the carbon cycle of the terrestrial Critical Zone, extending from the tree canopy to the aquifer, is crucial for accurate quantification of its total carbon storage and for modelling terrestrial carbon stock responses to climate change. Caves and their catchments offer a natural framework to sample and analyse carbon in unsaturated zone reservoirs across various spatial and temporal scales. In this study, we analyse the concentration, stable carbon isotopic ratio ($\delta^{13}C$), and radiocarbon ($^{14}C$) compositions of $CO_2$ from atmosphere, boreholes (0.5 to 5 m depth), and cave sampled every two months over two years at Milandre cave in northern Switzerland. High concentrations of up to 35'000 ppmV $CO_2$ are measured in the boreholes. The $\delta^{13}C$ values of $CO_2$ in the boreholes reflect the $\delta^{13}C$ of C3 plants (~ -26 ‰) which dominate the catchment ecosystem. Shallow meadow boreholes host older $CO_2$ in winter and modern $CO_2$ in summer, while forest ecosystems consistently export modern $CO_2$ ($F^{14}C$ = ~1) to the unsaturated zone. Cave $CO_2$ concentrations exceed atmospheric levels and are diluted by temperature-driven seasonal ventilation. Keeling plot intercepts indicate that the cave $CO_2$, which mixes with atmospheric $CO_2$, is younger in summer ($F^{14}C$ = 0.94) and older in winter ($F^{14}C$ = 0.88), with a $\delta^{13}C$ consistent with the C3 plant dominated catchment. Mixing models utilising drip water dissolved inorganic carbon $^{14}C$ suggest that varying carbonate dissolution and degassing dynamics do not explain the $F^{14}C$ variation and $\delta^{13}C$ stability of the mixing endmember. Rather, contributions from deeper aged carbon in the epikarst are likely. This study provides valuable insights into $CO_2$ source dynamics and cycling within karstic Critical Zones, highlighting the impact of seasonal variations and ecological factors on downward carbon export from terrestrial ecosystems.





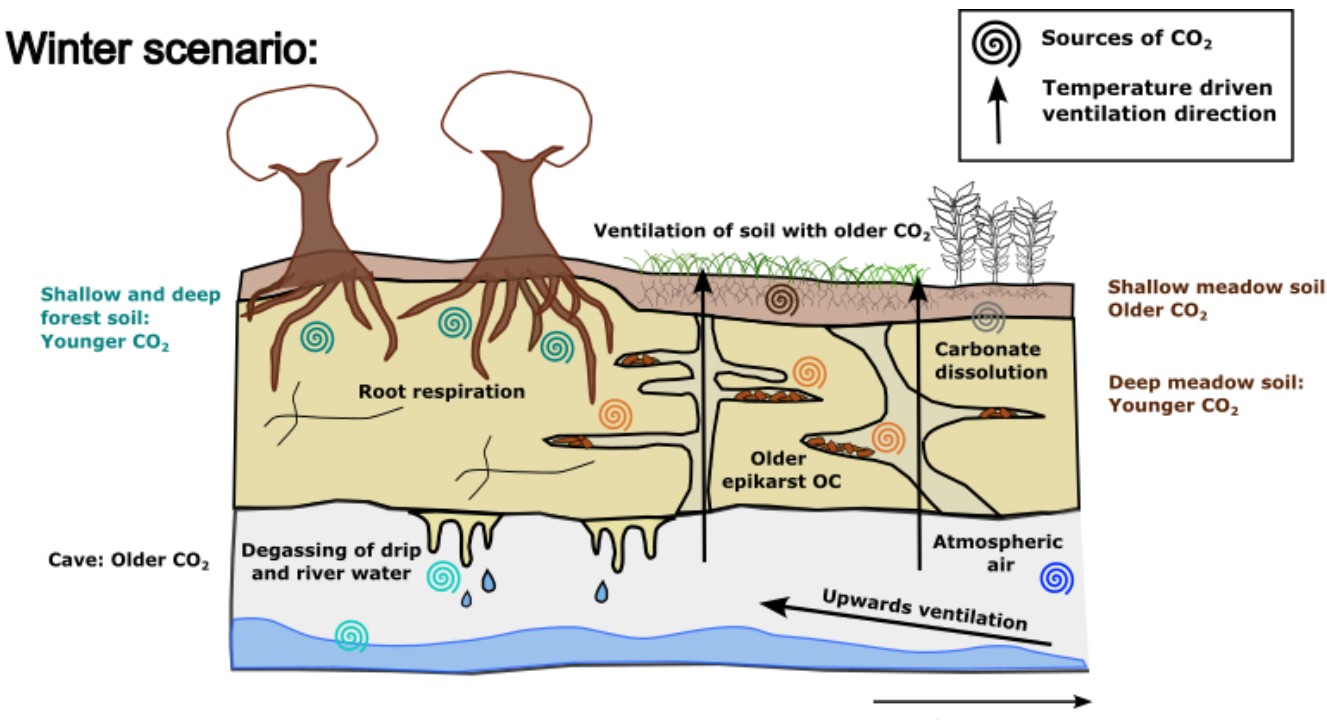

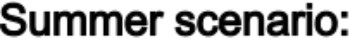

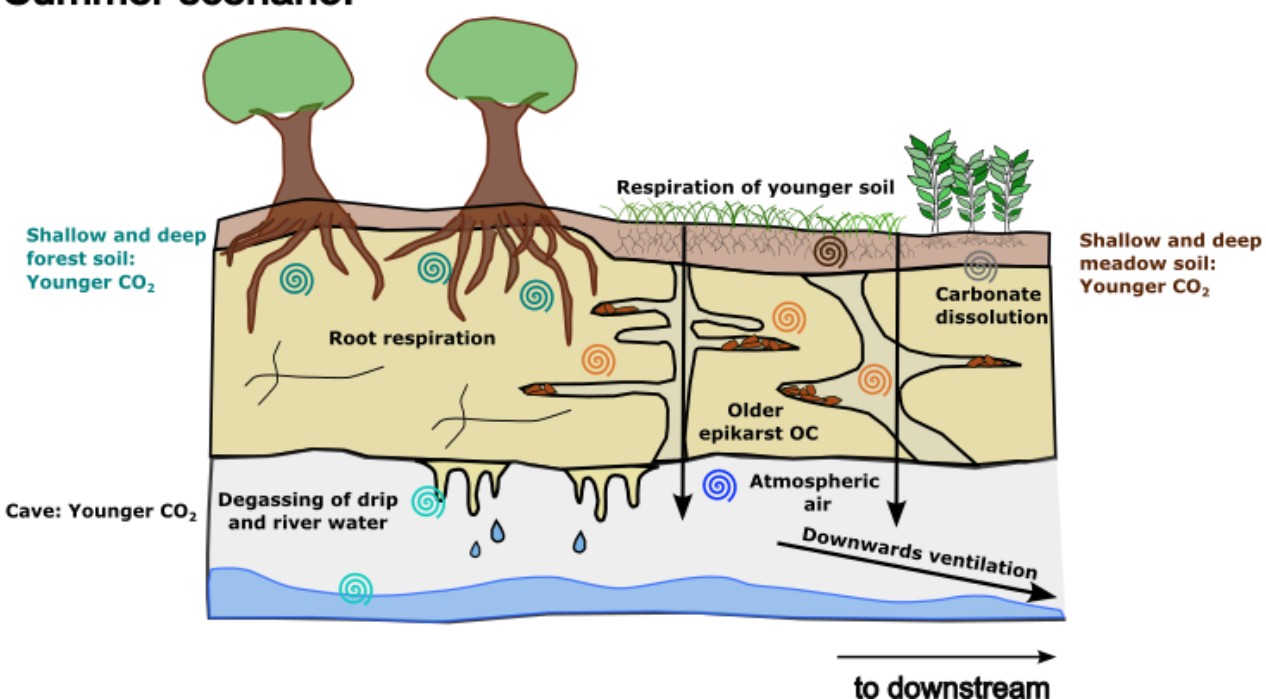



## 1 Introduction

Within the context of current and future changing climate, investigations into the response of Critical Zone carbon pools to rising temperatures and changing hydroclimatic conditions are crucial (Brantley et al, 2007). Recent studies have specifically highlighted the unsaturated zone as a potentially important but poorly constrained reservoir of gaseous $CO_2$ (Mattey et al.,

2016; Noronha et al., 2015; Keller, 2019; Stewart et al., 2022). Estimates suggest that between 2 to 53 PgC could be present in the form of $CO_2$ in the unsaturated zone globally (Baldini et al., 2018). This carbon is particularly vulnerable to changes in the water table level, whereby rises may easily result in the rapid release of $CO_2$ into the atmosphere (Baldini et al., 2018). Despite its importance, comprehensive assessments integrating spatial and temporal variability of shallow subsurface $CO_2$ remain scarce.

Understanding unsaturated zone $CO_2$ dynamics is complicated by the different sources of carbon contributing to the subsurface reservoir. The $CO_2$ present in the unsaturated zone is often referred to as ground air and was first defined as $CO_2$ produced by microbial oxidation of organic material which was transported from the surface (Atkinson, 1977). However, ground air can also refer more generally to high $CO_2$ concentrations in the subsurface, without linking its presence to a particular source.

Early evidence for the presence of a ground air reservoir was the observation of high $CO_2$ concentrations found in deep cave passages with poor connection to the catchment surface, suggesting an endogenous source of $CO_2$ (McDonough et al., 2016). The age profile of this high concentration $CO_2$ reservoir, however, varies by site, suggesting significant variability in its source and processing. Exclusively aged $CO_2$ reservoirs have been reported (Breecker et al., 2012; Noroha et al., 2015; Mattey et al., 2016; Bergel et al., 2017), as well as modern subsurface $CO_2$ pools, likely derived from ecosystem respiration of recently fixed

carbon (Campeau et al., 2019; Tune et al., 2020). Consequently, the primary sources of subsurface $CO_2$ generally considered are 1) the outside atmosphere (Kukuljan et al., 2021), 2) catchment soil and vegetation respiration (Breecker et al., 2012; Li et al., 2024), 3) microbial respiration in the unsaturated zone (Mattey et al., 2016; Ravn et al., 2020), 4) carbon dissolved from the carbonate bedrock (Milanolo & Gabrovšek, 2015), and 6) volcanic and metamorphic hydrothermal input (Chiodini et al., 2008; Girault et al., 2018).


Carbon isotope analysis of $CO_2$ can be used to differentiate the contributing sources to the subsurface gas mixture. Specifically, the $\delta^{13}C$ of $CO_2$ in the unsaturated zone can provide information about the influence of biological fractionation by overlying catchment vegetation and respiration by microorganisms, and the influence of carbonate dissolution and subsequent degassing (Breecker et al., 2017, McDonough et al., 2016). In addition, due to the contrast between the $^{14}C$ content of biospheric carbon,

which ranges from bomb peak enriched to >10'000 years old, and carbonate rock which is typically radiocarbon dead, $^{14}C$ measurement of subsurface $CO_2$ can provide information for both source apportionment and carbon turnover rates in the terrestrial subsurface (Noronha et al., 2015).





Though there has been extensive work investigating unsaturated zone $CO_2$ reservoirs, questions remain about the

concentration, composition and sources of ground air on temporal and spatial scales. This study seeks to address this gap by presenting a detailed assessment of $CO_2$ concentrations and isotopic compositions over 2 years within the Milandre cave karst system in Switzerland. We aim to 1) Determine the relationships between $pCO_2$, $\delta^{13}CO_2$, and $^{14}CO_2$ in Milandre cave and its catchment; 2) Assess the effect of seasonality on $CO_2$ concentration and isotopic characteristics and 3) Gain insights into the provenances of $CO_2$ and how these evolve over time.

**2 Site Overview**

Milandre cave (47.4852 ˚ N, 7.0161 ˚ E, 373m a.s.l.) is located in the municipality of Boncourt in the Jura canton, NW Switzerland (Fig. 1). The Milandre karst structure formed within the Jura mountains, a sub-alpine mountain range which lies laterally in a northwest-southeast direction. The cave is located within the external Plateau unit of the Jura mountains which consists of thin sub horizontal Mesozoic limestone units that have experienced deformation which produced imbricates and

tear faults (Sommaruga, 2011). More specifically, the Milandre karst system formed within the St-Ursanne Formation which overlies Oxfordian marls (Jeannin, 1998). The unsaturated zone ranges between 40 and 80 m depth, with a saturated zone of ~ 20 m (Perrin et al., 2003). The cave formed over 2.7 km, with 10.5 km of galleries forming along N-S oriented fractures. The Milandre river flows for 4.5 km in a northerly direction in the lower part of the cave and exits via the perennial Saivu spring where it joins the Allaine River, and the Bame temporary spring. The Milandre river drains a catchment area of

approximately 13 km² (Perrin et al., 2003). There is a series of dolines in the catchment area which form depressions of ~10 to 20 m in the landscape. Notably, one doline is located in the meadow partially covering the upstream section of Milandre cave.



**Figure 1. Map of the main passages of the Milandre cave network. The location of the atmospheric sampling (yellow), the soil (green) and epikarst boreholes (blue), and the cross-trip sampling (red) are annotated. Cave survey modified from Gigon & Wenger (1986). Base map from ©swisstopo.**




Over the past 40 years, land use in the catchment area has been dominated by farmland (37 %), forest (36 %), and meadows (12 %) as estimated by aerial photo analysis (Jeannin et al., 2016). The crops grown primarily include maize and tobacco, two C4 plants. Long-term land use analysis shows that farmland has decreased slightly in this area and special infrastructures have increased due to the construction of a motorway that overlies part of the cave. The soils of the area are influenced by the ecosystem that is overlying it. Forest soils are shallow leptosols (< 10 cm deep) that are rich in fragments of the carbonate bedrock. The shallow forest soils transition into deeper (up to ~ 80 cm) organic rich histosols in the meadows, particularly in the dolines.

The Jura region experiences a marine west coast, warm summer climate (Cfb classification) (Kottek et al., 2006). Daily temperature measurements from the Fahy MeteoSwiss weather station located ~ 9.5 km SW from Milandre cave show an average temperature of 9.4 °C (1991 to 2020). Temperature seasonality is strong, with monthly average temperatures fluctuating between a minimum of -1.2 °C in January and a maximum of 18.2 °C in July from 1991 to 2020 (MeteoSwiss, 2024). In contrast, temperatures within the cave remain almost constant year-round, and vary between 10.3-11.0 °C (Affolter et al., 2020). The temperature difference between the outside and in-cave temperature drives dynamic ventilation on seasonal scales (Garagnon et al., 2022). Regional meteoric precipitation shows an average of 1046 mm year$^{-1}$. Monthly precipitation data indicate that precipitation in this area is well dispersed throughout the year (MeteoSwiss, 2024). Monitoring of tritium ($^3$H) in stalagmite drip water was used to estimate the residence time of seepage water at ~ 5.5 to 6.6 years (Affolter et al., 2020).

## 3 Materials and methods

### 3.1 Gas sampling set up

All gas samples were taken every two months from December 2021 to January 2024. The samples were collected in 5 L sampling bags (Cali-5-Bond, Calibrated Instruments, USA) using a handheld pump. The gas was dried through a glass tube filled with granular magnesium perchlorate (~ 83 % purity, Supelco, Germany) to reduce the effects of humidity which may affect the isotopic composition of the sample. The sampling set up and procedure were identical for all gas samples. The $CO_2$ concentration was monitored during line flushing using an in-house built NDIR $CO_2$ sensor (SCD30, Sensirion, Switzerland) to ensure accurate sampling. The magnesium perchlorate was exchanged before each sampling day. Prior to analysis, the samples were stored away from direct sunlight in cool temperatures for a maximum of six weeks. The same bags were reused for each sampling location and were flushed with $N_2$ gas for at least 48 h before sampling to reduce cross contamination risks.

Atmospheric samples were collected at a defined sampling site above the cave (Fig. 1). Before sampling, the line was manually flushed for 1 min. To reduce the risk of breath contamination, the sampling set up was attached to 3 m long Teflon tubes, and samples were always taken against air flow direction.




The unsaturated zone air was sampled from six boreholes of varying depths between 0.5 and 5 m (Table 1). Spatially, the boreholes cover a large portion of the cave's upstream hydrological catchment, and are overlain by contrasting vegetation cover (mixed-deciduous forest or a grass meadow) (Fig. 1) (Table 1). The gas was sampled in two lines from depths of 0.5 to 0.85 m (Shallow 1), and three lines from 0.6 to 1.5 m (Shallow 2), and from deeper depths in single lines of 5 m (Deep 1, Deep 2,

Deep 3, Deep 4). Due to the nature of the installation, it was not possible to assign a specific depth to the multi-line boreholes. All boreholes were drilled in 2013 and are equipped with aluminium tubes compacted by layers of gravel, bentonite and sand. The sampling line was flushed for 1 min once attached to the borehole, and then samples were taken into the Cali-5-Bond bags. Due to the nature of the borehole installation, we assume that the samples were not taken in steady state conditions, where the production rate of $CO_2$ would equal the sampling rate.


| ID | Well names | SISKA ID | Easting (CH1903+/LV95) (Swiss grid) | Northing (CH1903+/ LV95) (Swiss grid) | Depth (m) | Number of sampling lines | Classification | Ecosystem cover |
|---|---|---|---|---|---|---|---|---|
| SG-04 | Shallow 1 | CO2-13 | 2'567'083 | 1'256'904 | 0.5 to 0.85 | 2 | Soil | Forest |
| SG-07 | Shallow 2 | CO2-10 | 2'567'074 | 1'257'068 | 0.6 to 1.5 | 3 | Soil | Meadow |
| SG-05 | Deep 1 | EPI-2 | 2'567'097 | 1'256'896 | 5 | 1 | Epikarst | Forest |
| SG-10 | Deep 2 | EPI-4 | 2'567'089 | 1'256'890 | 5 | 1 | Epikarst | Forest |
| SG-08 | Deep 3 | EPI-7 | 2'567'080 | 1'257'063 | 5 | 1 | Epikarst | Meadow |
| SG-09 | Deep 4 | EPI-5 | 2'567'096 | 1'257'059 | 5 | 1 | Epikarst | Meadow |

**Table 1 Summary of the boreholes sampled in Milandre cave catchment notating sample depth, number of sampling lines, classification in soil/ epikarst zone, and ecosystem cover type. The ID's are the names given to the boreholes for this study, and are concurrent with the supplementary data provided. SISKA (Swiss Institute for Speleology and Karst Studies) IDs refer to the original**
**names of the boreholes at the time of installation.**

Cave air was sampled in two locations within Milandre cave proximal to both entrances, Downstream (topographically lower) and Upstream (topographically higher), with the same procedure as for the atmospheric samples (Fig. 1). To gain insight into possible spatial variability within the cave, a cross-trip through the main passage of the cave was carried out in September 2021, and 14 samples were collected (Fig. 1). The trip was taken against the direction of air flow, from the downstream to

upstream entrance, again to reduce risk of breath contamination.



## 3.2 Continuous O₂ and CO₂ concentration monitoring

Soil gas $O_2$ and $CO_2$ concentrations were measured continuously nearby Shallow 2 at 40 cm depth and recorded at 10 min intervals between April 2023 and June 2024 using a SCD-41 $CO_2$ sensor (± 50 ppmV + 5 % of reading) (Sensirion, Switzerland) and SGX-40X $O_2$ sensor (Amphenol SGX, Sensortech, Switzerland). Calibration was cross-checked with an independent, handheld multi-gas monitor (XAM 5600, Dräger, Germany) before installation.

## 3.3 CO₂ concentration and δ¹³C analysis

The concentration of $CO_2$ and the stable carbon isotopic composition ($\delta^{13}C$) in all gas samples was measured using a cavity ringdown spectrometer (CRDS) G2131-I Isotopic $CO_2$ instrument (Picarro, USA) at ETH Zurich ($CO_2$ concentration = 0.2 ppm, $\delta^{13}C$ = < 0.1 ‰ precision). Standard gases with known $CO_2$ concentrations, 399.6 ppmV and 2000 ppmV $CO_2$ in synthetic air, were measured in addition to standard gases of known $\delta^{13}CO_2$ values, -27.8 ‰ and -2.8 ‰ VPDB, for offline calibration. As there was no standard gas available for similarly high $CO_2$ concentrations as in some samples, linearity of the concentration data produced by the CRDS is assumed. All standards and samples were measured by directly attaching the sample bags to the inlet of the CRDS that was previously fitted with a magnesium perchlorate dryer, and measuring gas concentration and isotopic composition for 2 min after reaching steady state. For the evaluation, we used the mean and standard deviation of this measurement interval. $\delta^{13}C$ refers to the ratio between the two stable carbon isotopes with respect to the Vienna Pee Dee Belemnite (VPDB) standard:

$$\delta^{13}C\ (‰)\ =\ \left(\frac{^{13}C/^{12}C_{sample}}{^{13}C/^{12}C_{standard}} - 1\right) \times 1000 \qquad (1)$$

## 3.4 Radiocarbon analysis

The $CO_2$ from each gas sample was converted into approximately 1 mg of graphite using an Automatic Graphitization Equipment (AGE, IonPlus, Switzerland) (Wacker et al., 2010a) coupled with a custom-made carbon inlet system. In this process, each sampling bag was successively attached to the inlet system which dried the sample through a fine-grained magnesium perchlorate water trap via a vacuum line. The volume of air required to sample was calculated depending on the $CO_2$ concentration of the sample, and the appropriate amount was trapped within a stainless-steel coil that was cooled by submersion in a liquid $N_2$ bath. After trapping, the $CO_2$ was sublimated by submerging the coil in a room temperature water bath and then adsorbed onto a zeolite trap using helium as a carrier gas between the coil and the trap. The $CO_2$ adsorbed onto the trap was then thermally desorbed and filled into the AGE reactor, where it was reduced to graphite with hydrogen over an iron catalyst (Wacker et al., 2010a). Oxalic Acid II gas ($F^{14}C$ = 1.3407, Oxa II, NIST SRM 4990C, HEKAL AMS Lab, Hungary) as a modern standard and a radiocarbon fossil reference $CO_2$ gas ($F^{14}C$ = 0, $CO_2 \geq$ 99.7 % Vol.abs, Carbagas, Switzerland) were also graphitized and used to calibrate the measurements. The resulting graphite was pressed into targets for



radiocarbon analysis through AMS analysis with a MIni CArbon DAting System (MICADAS, Ionplus, Switzerland) (Synal et al., 2007; Szidat, 2020). The resultant radiocarbon data were corrected using the BATS (4.3) software (Wacker et al., 2010b). In this study the $^{14}$C content of samples will be discussed in $F^{14}$C notation according to Reimer et al., (2004):

$$F^{14}C = \frac{^{14}C/^{12}C_{sample}}{^{14}C/^{12}C_{standard}} \qquad (2)$$

**3.5 Dissolved inorganic carbon analysis**

Drip water dissolved inorganic carbon (DIC) $^{14}$C was measured to constrain the DIC degassing endmember. Samples were measured over one year from December 2021 and December 2022 at 9 drip sites focused on the actively dripping galleries nearby Upstream and Downstream (Fig. 1). Using a 5 mL syringe, 1 mL of drip water was collected directly from the soda straw stalactites on the cave roof to reduce fractionation effects associated with degassing. The water was injected directly into pre-cleaned Exetainer® (Round Bottom, Borosilicate, 938W, Labco Limited, UK) vials which had been flushed with helium gas and pre-spiked with 150 µL of 85 % $H_3PO_4$ (Suprapur®, 85 %, Merck KGaA, Germany). The $^{14}$C content of the DIC was measured by sampling the vial headspace using the Carbonate Handling System (CHS, Ionplus, Switzerland) and transferring the gaseous sample to the AMS using a gas handling system. Here, the $CO_2$ was introduced to the gas ion source through the CHS at a flow of 50 mL min$^{-1}$. The $^{14}$C content for each sample was measured for ~ 60 cycles. Standard material with a carbonate matrix of similar composition to the DIC samples were used as standards (IAEA C1 ($F^{14}$C = 0) and C2 ($F^{14}$C = 0.411), sodium bicarbonate (NaHCO$_3$, Sigma Aldrich, USA) and potassium bicarbonate (KHCO$_3$, Sigma Aldrich, USA) (both $F^{14}$C = 0)).

**3.6 Data analysis and statistics**

All data and statistical analyses and all graphs were generated using the Python 3 programming language (Van Rossum & Drake, 2009).

Relationships between $CO_2$ concentration, $\delta^{13}$C, $F^{14}$C for each sample type and the Mean Monthly Temperature (MMT) and Mean Monthly Precipitation (MMP) from Fahy weather station were explored using Spearman's rank correlation coefficient analysis (Kokoska & Zwillinger, 2000). A significantly correlated relationship between two variables is defined by $p < 0.05$, with the correlation coefficient denoted by "ρ", and number of samples involved in the analysis as "n".



# 4 Results

## 4.1 Atmospheric CO₂

The atmospheric samples had a $CO_2$ concentration ranging from ~ 380 ppmV (August 2023) to ~ 485 ppmV (December 2022) with an average of ~ 440 ppmV. The $\delta^{13}C$ ranged from -12.5 ‰ (February 2023) to -7.6 ‰ (August 2023) and had a mean of -10.1 ‰ (n = 17). The $CO_2$ sampled was typically modern, though had some fluctuation from $F^{14}C$ 0.98 (December 2022) to 1.02 (June 2022) and a mean of $F^{14}C$ 1.0 (Fig. 2).

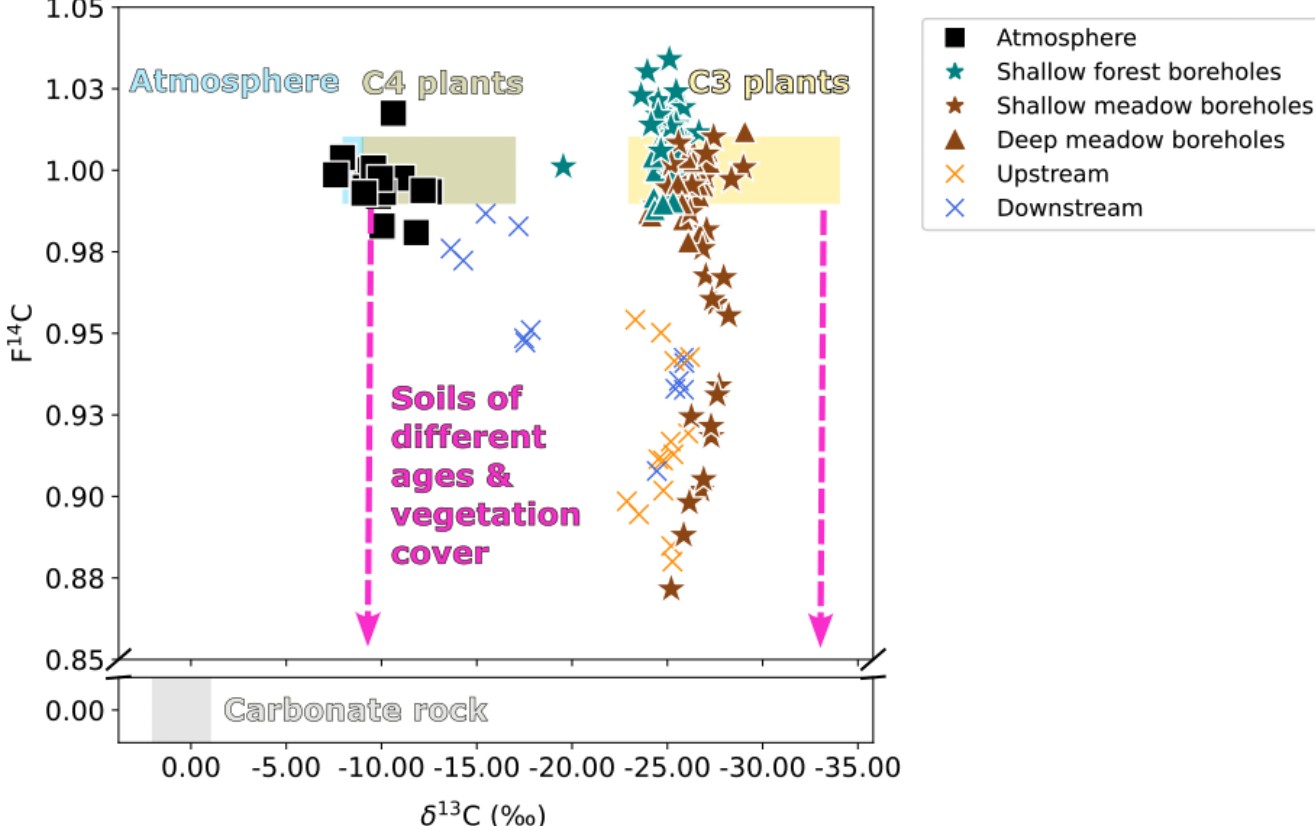

**Figure 2. $\delta^{13}C$ and $F^{14}C$ of all gas samples over the monitoring period. The theoretical carbon reservoirs present in the system are shown with the atmosphere (blue), C3 plants (yellow), C4 plants (green), soils of various ages and compositions (pink arrows), and carbonate rock (grey).**





## 4.2 Soil Zone $CO_2$

220

Soil boreholes have higher $CO_2$ concentrations and lower $\delta^{13}C$ compared to the atmosphere (Fig. 2). Concentrations in boreholes that sample the soil zone (Shallow 1 and Shallow 2) (Fig. 3a) vary significantly depending on sampling depth, cover type, and season. The Shallow 1 boreholes (forest) show seasonal high $pCO_2$ in the summer months from June to October, which steadily declines during winter until it begins to rise again after April. The Shallow 1 (forest) boreholes vary from a

225 maximum of ~ 18'000 ppmV in August 2022 to a minimum in January 2024 of ~ 1200 ppmV, with an average of ~10'700 ppmV (n = 21). The Shallow meadow boreholes range from ~ 27'000 ppmV in June 2023 to ~ 4800 ppmV in February 2023 with an average of ~ 9900 ppmV (n = 30).

The $\delta^{13}C$ varies by 7.0 ‰ in shallow forest boreholes over the sampling period ranging from a minimum of -26.7 ‰ in August

230 2023 to a maximum of -19.5 ‰ in January 2024 (Fig. 3b). There is 4.0 ‰ isotopic variation in the shallow meadow boreholes with a minimum of -29.0 ‰ in October 2022 and a maximum of -24.4 ‰ in August 2022. The $pCO_2$ and $\delta^{13}C$ are negatively correlated in both shallow forest and shallow meadow boreholes (forest: $\rho = -0.47$, $p = 0.02$, n = 25. meadow: $\rho = -0.49$, $p = 0.01$, n = 30), and $\delta^{13}C$ is on average lower in shallow meadow than shallow forest boreholes.





**Figure 3. Maximum and minimum a) CO₂ concentration, b) δ¹³C and c) F¹⁴C of soil borehole samples from Shallow 1 in the forest (green) and Shallow 2 in the meadow (brown). Summer (yellow) and winter (blue) seasons are highlighted by the coloured background. Concentrations are shown as ranges between maxima and minima as they aggregate over several sampling lines in the soil per location (Table 1).**

235

240



F¹⁴C in Shallow 1 boreholes (forest) shows little annual variability and fluctuates slightly around a modern value (max = 1.03 in April 2022, min = 1.0 in June 2023) (Fig. 3c). Conversely, the shallow meadow boreholes show distinct annual variation in F¹⁴C, which steadily decreases from modern (1.0) in August 2022 and 2023 to a low in April 2023 and a minimum in January 2024 (0.87). The samples from the shallow forest boreholes have a positive correlation between F¹⁴C and $\delta^{13}C$ ($\rho$ = 0.48, p = 0.02, n = 25). In the shallow meadow borehole samples, we observe a positive correlation between F¹⁴C and MMT ($\rho$ = 0.67, p = 0.001, n = 30). All other tested parameters show no significant correlations (see Appendix A).

The continuously measured $O_2$ and $CO_2$ concentrations nearby Shallow 2 in the meadow show an inverse relationship (Fig. 4). The concentrations are relatively stable throughout most of the year with higher $O_2$ and lower $CO_2$ concentrations. This pattern is disturbed by a sharp decrease in $O_2$ and an increase in $CO_2$ observed around May/June of both 2023 and 2024. These different conditions last around one month before returning to the typical concentrations. The $O_2$ concentration ranges from a maximum of 20.2 % in August 2023 to a minimum of 17.8 % during a brief interval in May 2023. Conversely, the $CO_2$ concentration peaks at ~ 17'000 ppmV in May 2023 and is the lowest at ~ 2500 ppmV in August 2023. The Shallow 2 boreholes also show peaks in $CO_2$ concentration in May to June 2023 (Fig. 4).

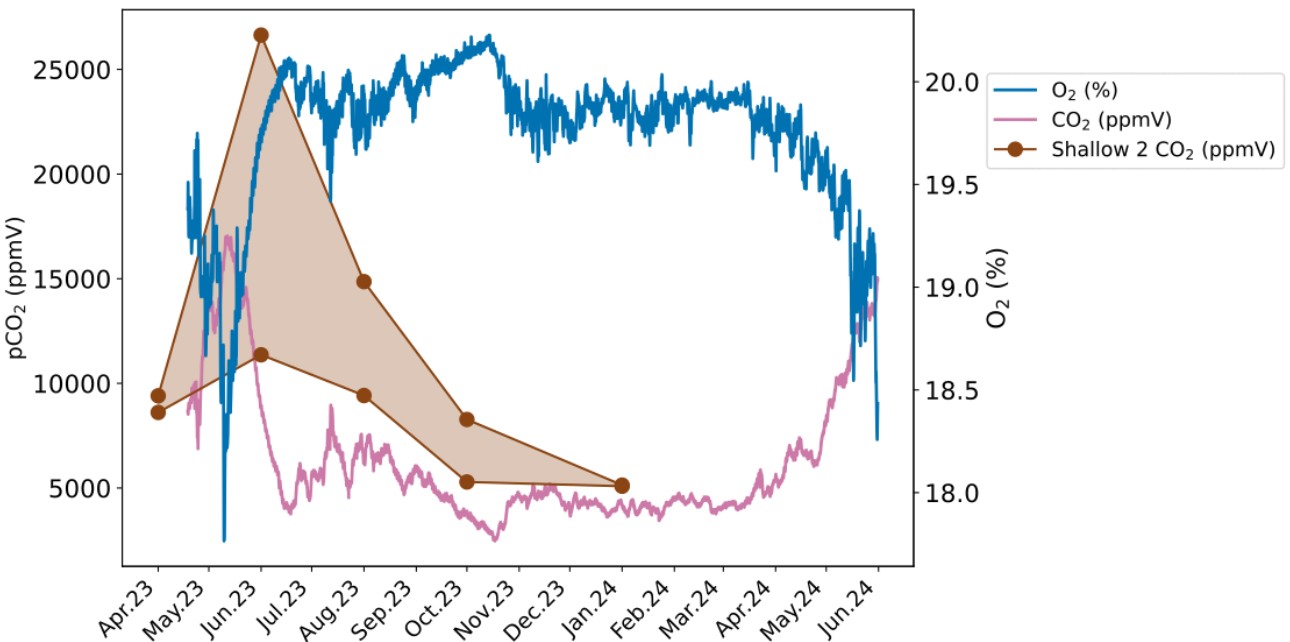

**Figure 4. Continuously measured CO₂ (pink) and O₂ (blue) concentrations from the soil zone nearby the Shallow 2 boreholes. CO₂ concentrations from Shallow 2 are also plotted (brown).**



### 4.3 Epikarst CO₂

Epikarst CO₂ concentrations varied widely during the sampling period but generally are very high (>10'000 ppmV). On average, the forest boreholes Deep 1 and Deep 2 have higher concentrations than the meadow boreholes Deep 3 and Deep 4.

Concentrations are generally highest in the Deep 1 borehole (forest) (Fig. 5a) with a maximum of ~ 37'000 ppmV in June 2023 and a similarly high concentration peak in June 2022. The lowest values (~ 21'000 ppmV) at this borehole are recorded in the winter months. Deep 2 (forest) shows an opposite trend with its highest CO₂ concentrations in autumn, winter, and spring (max = ~ 29'000 ppmV in December 2022). The lowest CO₂ concentrations were measured during the summer months (min = ~ 7900 ppmV in June 2023). There is comparatively less variability in the Deep 3 and Deep 4 boreholes (both in the meadow).

In Deep 3, concentrations fluctuate from ~ 6100 to ~ 19'000 ppmV with no obvious seasonality. Deep 4 has higher concentrations over the winter period and lower during summer, peaking at ~ 14'000 ppmV in October 2022 and dropping to an overall minimum of ~ 3000 ppmV in June 2023.

The δ¹³C of the majority of epikarst CO₂ samples is very stable around ~ -26 ‰ (Fig. 5b). Similar to the shallow boreholes,
samples from deep meadow boreholes (Deep 3 and 4) have on average slightly lower δ¹³C values. In the meadow boreholes Deep 3 and Deep 4, the δ¹³C and pCO₂ are negatively correlated (ρ = -0.66, p = 0.001, n = 21) (see Appendix A).



**Figure 5. a) CO₂ concentration, b) δ¹³C and c) F¹⁴C of epikarst borehole samples from forest boreholes Deep 1 and Deep 2 (green), and meadow boreholes Deep 3 and Deep 4 (brown). Summer (yellow) and winter (blue) seasons are highlighted by the coloured background.**

The F¹⁴C of epikarst CO₂ samples is relatively stable throughout the sampling period, ranging from 1.01 (Deep 1, June 2022) to 0.98 (Deep 4, October 2022) (Fig. 5c). A possible seasonal pattern can be observed in the meadow boreholes (Deep 3 and Deep 4) with lower F¹⁴C values in the winter and spring months and higher F¹⁴C during summer, similar to the observation from the shallow meadow boreholes. No clear seasonal pattern is observed for Deep 1 and Deep 2. The F¹⁴C epikarst CO₂ samples are negatively correlated with their δ¹³C value (ρ = -0.54, p = 0.01, n = 21) and positively correlated with MMT (ρ = 0.53, p = 0.01, n = 21). All other tested parameters show no significant correlations.





## 4.4 Cave CO₂

The $CO_2$ concentrations at the Downstream and Upstream sampling points in the cave are inversely correlated (Fig. 6a). Downstream shows higher $pCO_2$ from June to October in 2022 and 2023, with a maximum of ~ 14'000 ppmV in August 2023. The lowest concentrations are seen between December and April, reaching close to atmospheric concentrations with a minimum of ~ 420 ppmV in December 2021. Conversely, the highest $CO_2$ concentrations at the Upstream site were measured during the winter months, with a maximum of ~ 30'000 ppmV in December 2021. The lowest $CO_2$ concentrations at Upstream

were observed during the summer months with a minimum of ~ 2300 ppmV in August 2022. Overall, concentrations at the Upstream site reach higher maxima and do not approach atmospheric values at their minimum, as at the Downstream site. The isotopic composition of $CO_2$ is inversely related to its concentration, with higher $CO_2$ concentration coinciding with lower $F^{14}C$ and $\delta^{13}C$ values. Seasonal trends differ between sites, at the Downstream site $\delta^{13}C$ and $F^{14}C$ values are higher in winter and lower in summer, whereas at the Upstream site, $\delta^{13}C$ and $F^{14}C$ values show the opposite patten (Fig. 6b and c). The $\delta^{13}C$

in Downstream ranged between -25.9 ‰ in June 2023 and -13.6 ‰ in December 2021, and the $F^{14}C$ between 0.91 in October 2023 and 0.98 in February 2023. At Upstream, the opposite relation with $CO_2$ concentration is well expressed for $F^{14}C$, but not as clear for $\delta^{13}C$, with minimal variability over the entire study period. The $\delta^{13}C$ in Upstream fluctuates slightly around a mean of -24.8 ‰.




**Figure 6. a) CO₂ concentration, b) δ¹³C and c) F¹⁴C of cave air samples from the Downstream (red) and Upstream (orange) sites. Summer (yellow) and winter (blue) seasons are highlighted by the coloured background. The approximate timing of the temperature driven ventilation direction changes are shown by the annotations.**



The $F^{14}C$ of $CO_2$ in Upstream is overall lower than that of Downstream, increasing over late spring 2022 and into summer, peaking during August 2022 at 0.95. Decreasing $F^{14}C$ is observed during winter into spring with lows of ~ 0.88 in February 2023. At the Downstream site, the $pCO_2$ and $\delta^{13}C$ ($\rho$ = -0.9, p = 0.0001, n = 14), $pCO_2$ and $F^{14}C$ ($\rho$ = -0.83, p = 0, n = 14), and $\delta^{13}C$ and MMT ($\rho$ = -0.79, p = 0.0001, n = 14) are negatively correlated. Moreover, the $pCO_2$ and MMT ($\rho$ = 0.53, p = 0.05, n = 14), and $\delta^{13}C$ and $F^{14}C$ ($\rho$ = 0.74, p = 0, n = 14) are positively correlated. At the Upstream site, $pCO_2$ and MMP are positively

correlated ($\rho$ = 0.55, p = 0.04, n = 14), and $pCO_2$ and MMT are negatively correlated ($\rho$ = -0.54, p = 0.05, n = 14) (See Appendix A). All other tested parameters show no significant correlations.

   To constrain the isotopic composition of the end members contributing $CO_2$ to the cave air mixture, we use the Keeling plot approach (Fig. 7a & b). This approach assumes that cave air is a mixture of two main sources, the atmosphere (with known

concentration and isotopic values), and a second source of a priori unknown composition. The y-intercepts of the Keeling plots represent the isotopic composition of the contributing end member which mixes with atmospheric air inside the cave (Keeling, 1961; Pataki et al., 2003) (Fig. 7a & b). We find that over time, the isotopic composition of the endmember varies for both $\delta^{13}C$ and $F^{14}C$ (Fig. 7c). The endmember $\delta^{13}C$ value varies slightly by ~ 2.0 ‰ from -26.8 ‰ in June 2023 to -25.0 ‰ in February 2022. The variation in the $F^{14}C$ is larger, ranging from 0.88 in February 2023 to 0.94 in June 2022, August 2022,

October 2022, and June 2023 (ca. ~ 0.06 $F^{14}C$). Maxima in the $F^{14}C$ value derived for the endmember through the Keeling plot generally correspond to decreases in $\delta^{13}C$, with two $F^{14}C$ maxima occurring in June to October 2022 and in June to August 2023.





**Figure 7. Keeling plots for δ¹³C (a) and F¹⁴C (b) of cave (Upstream and Downstream sites) and atmospheric samples from all monitored months. The dashed regression lines denote the months with the maximum and minimum y-intercepts for δ¹³C and F¹⁴C. c) Keeling plot y-intercepts for δ¹³C (blue) and F¹⁴C (orange) over time. Summer (yellow) and winter (blue) seasons are highlighted by the coloured background.**

Samples taken along the main passage of the cave during the cross-trip show increasing $pCO_2$ from the lower entrance (location 1, corresponding to Downstream monitoring site, ~ 6900 ppmV) to a plateau of ~ 14'000 ppmV from sample point 4 to 9 towards the centre of the cave (Fig. 8a). The $pCO_2$ decreases sharply after sample point 9 until point 11 (~ 6600 ppmV). The $pCO_2$ then increases briefly at sample point 12 and then decreases again upon approach to the cave exit (Upstream). The δ¹³C values remain essentially constant at -25.5 ‰ through the initial segments of the cave, beginning to increase slightly past sample point 9 (Fig. 8b). The δ¹³C value is -25.8 ‰ at sample point 9 and progressively increases as $pCO_2$ levels drop between sampling points 9 and 11, followed by a strong increase toward the Upstream exit (-11 ‰). The F¹⁴C values are constant at ~ 0.93 between sample points 1 and 5, followed by more variability and generally lower F¹⁴C values in the middle of the transect,




coinciding with the river passage of the cave (minimum 0.89 at sampling point 11; Fig. 9c). Towards the Upstream exit, $F^{14}C$

begins to increase strongly (maximum 0.98 at sampling point 15; Fig. 8c). Sample 13 is excluded due to a ruptured bag during

sampling.

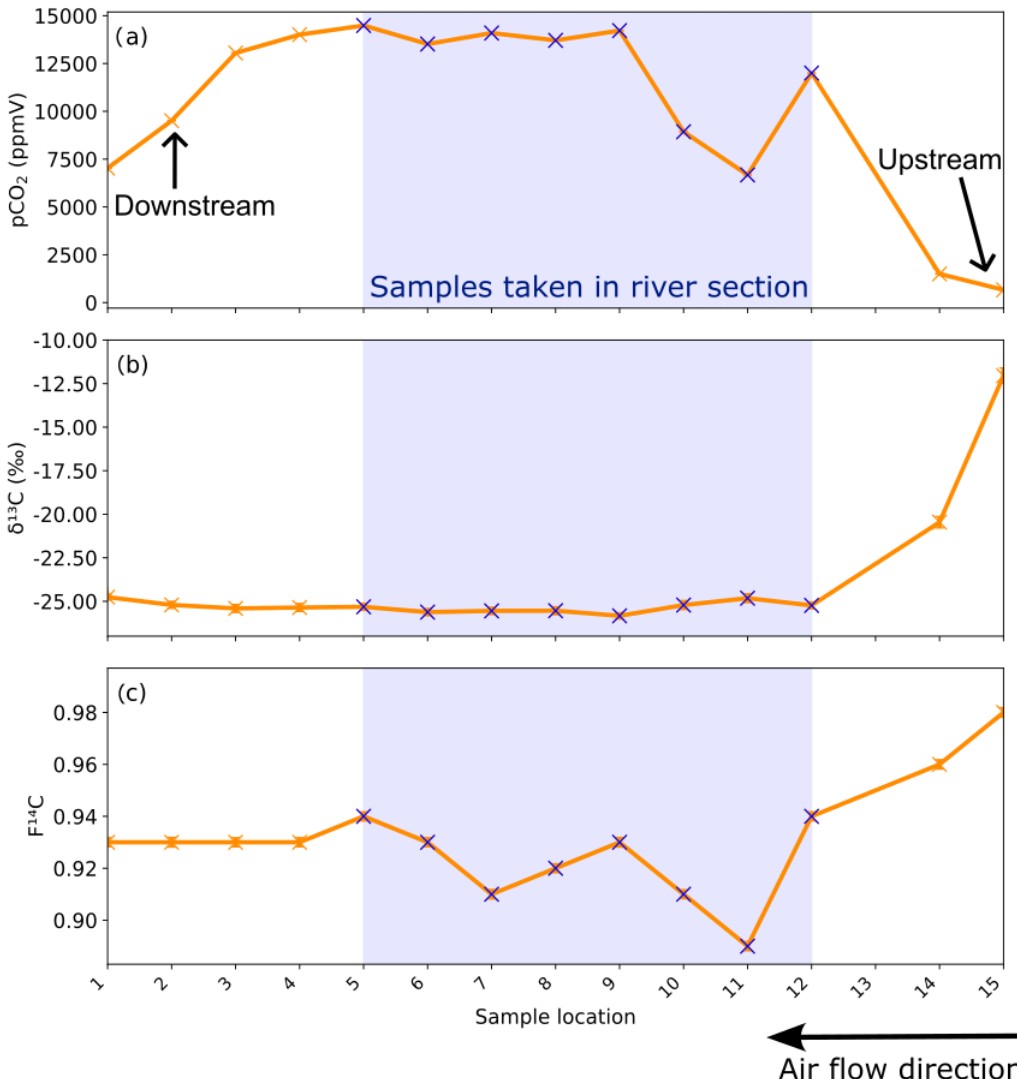

**Figure 8. a) $CO_2$ concentration, b) $\delta^{13}C$ and c) $F^{14}C$ of cave air samples from the different sites sampled during the cross trip.**
**Sampling locations are shown in Fig. 1. The blue crosses and blue bar notate where samples were taken standing in the river. The direction of cave air ventilation is shown by the lower arrow. The approximate locations of the Upstream and Downstream sampling sites are annotated.**





## 5 Discussion

### 5.1 Sources and variability of shallow depth CO$_2$

The observed fluctuations in CO$_2$ concentrations in the Shallow boreholes (0.5 to 1.5 m) varying with depth, vegetative cover, and season suggest complex carbon cycling dynamics taking place in the soil zone and shallow fractured epikarst. Overall, both samples from forest and meadow locations have similar average CO$_2$ concentrations. In the Shallow 1 forest boreholes, higher CO$_2$ concentrations occur during summer months (Fig. 3a), aligning with increased autotrophic (occurring in plant roots, leaves, and stems) and heterotrophic (microbial) respiration rates during warmer seasons resulting in higher CO$_2$

production compared to losses due to soil efflux to the atmosphere and downward transport in gaseous or dissolved form. Due to our discrete sampling approach, we cannot discern between higher production and increased losses. The decline in CO$_2$ concentrations during winter indicates reduced soil microbial respiration, and reduced autotrophic respiration, in response to lower temperatures. Though CO$_2$ production is reduced in winter, concentrations of several ~1000 ppmV are measured at our sites, perhaps due to the ability of soil bacteria to survive in below freezing temperatures as low as -5 to -7.5 ˚C (Kähkönen et

al., 2001), to persisting autotrophic respiration of shallow roots, or because of reduced transportation of CO$_2$ due to higher soil water content (Hashimoto & Komatsu, 2006). Similar seasonal trends in CO$_2$ concentrations have been observed in several other studies (Billings et al; 1998; Pumpanen et al., 2003; Zhang et al., 2023). Comparatively, CO$_2$ concentrations in the Shallow 2 meadow boreholes do not show as pronounced seasonality due to consistent CO$_2$ accumulation year-round.

δ¹³C of CO$_2$ in shallow boreholes located in both the meadow and forest correspond to the typical isotopic signature of C3 plants which dominate the catchment area, supporting our interpretation of a dominant biogenic source of carbon in the soil gas (Fig. 2) (Fig. 3b). Plants using the C3 metabolic pathway (e.g. most temperate vegetation) produce carbon with a δ¹³C of ~ -23 to -34 ‰ (Staddon et al., 2004). Furthermore, due to the nature of the sampling, the soil was not sampled in steady state conditions, potentially disturbing any existing CO$_2$ trends with depth. Though both meadow and forest boreholes have C3 plant

δ¹³C signatures, the δ¹³C of the meadow soils are on average lower than those in the forest soils. The meadow and forested areas have developed contrasting soil compositions, with the meadow soils deeper, more compacted, and with a higher organic content, whilst the forest soils are shallow, unconsolidated, and with more roots. Comparisons of root free and root containing soils have suggested that root respiration contributes lower δ¹³CO$_2$ to the soil gas (Diao et al., 2022). This is the opposite trend to what our findings would suggest. Efforts to disentangle the δ¹³C of autotrophic and heterotrophic contributions to soil CO$_2$

have shown a range of results, where autotrophic respiration can produce the same (Wu et al., 2017), higher (Moyes et al., 2010), or lower (Risk et al., 2012) δ¹³C than heterotrophic respiration. However, different soil compositions can affect soil moisture, which can cause both increases (Unger et al., 2010), decreases (Powers et al., 2010), and no change (Diao et al., 2022) in soil δ¹³C. It is therefore difficult in the context of this study to determine the exact reason for the difference in δ¹³C between surface covers. Interestingly, we also do not find evidence for contributions to subsurface gas from C4 plants, possibly



due to the absence of sampling locations in the fields in the catchment where C4 crops are grown, suggesting limited lateral gas transport.

The $CO_2$ $F^{14}C$ of meadow boreholes in Shallow 2 shows statistically significant seasonal behaviour ($\rho$ = 0.67, p = 0.001, n = 30), with more aged $CO_2$ dominating the winter and spring months ($F^{14}C$ ~ 0.88), and modern $CO_2$ in summer ($F^{14}C$ ~ 1.00)

(Fig. 3c). These seasonal shifts suggest a potential influence of temperature-sensitive processes on soil carbon dynamics, with higher autotrophic respiration rates in the warm months, as well as heterotrophic respiration of very recently fixed photosynthates dominating (Campeau et al., 2019). Though typically the productive growing season for meadow grasses occurs in spring (Wingler & Hennessy, 2016), there is a lag in the response in the $F^{14}C$, which peaks between June and October. This may be due to the time it takes for enough modern $CO_2$ to accumulate in the soil and dominating the average gas age. Similar

impacts of seasonality on the contributions of young carbon pools to the soil $CO_2$ budget have been observed in soils of various environments whereby autotrophic respiration and heterotrophic decomposition of younger carbon dominate in warmer months (Trumbore, 2000; Chiti et al., 2011; Vaughn & Torn, 2018). A contribution of fossil carbon from the dissolution of carbonate bedrock fragments in the soil zone could also be considered to explain the low soil $CO_2$ $F^{14}C$ values in winter. However, such a bedrock contribution would result in a concomitant increase in $\delta^{13}C$, as marine carbonates, which typically constitute the

karstified host rock, are characterized by $\delta^{13}C$ of values of ~ 0 ‰ (Planavsky et al., 2015) and up to + 2 ‰ in the region of Milandre cave (Weissert & Mohr, 1996) (Fig. 2), with a gaseous $\delta^{13}C$ ~ 9.54 ‰ lighter when at equilibrium with DIC from carbonate dissolution at 10.5 ˚C (Mook et al., 1974). Since we do not observe increases in $\delta^{13}C$ of soil gas during the periods of lower $F^{14}C$, this is an unlikely scenario. A further explanation for the seasonal variability could be due to the seasonal ventilation dynamics in the karst system, which promote upwards ventilation of cave air during the winter, and downwards

ventilation of soil air in summer (see section 5.4).

The modern $F^{14}C$ $CO_2$ signature in Shallow 1 forest boreholes (Fig. 3c) suggests a year-round dominant source of $CO_2$ from root respiration and the decomposition of very recently fixed soil organic matter. As the root systems are deeper and more developed in the forest soils, there is a higher input of modern carbon compared to the meadows. The soils in the forested area

are shallow (in some areas < 10 cm deep) and are dominated by large pebbles and fragments of the carbonate bedrock and shallow roots. Thus, shallow depth carbon sequestration may be suppressed in the forest soil, resulting in a $CO_2$ profile dominated by deeper root respiration (Hasenmueller et al., 2017; Tune et al., 2020).

## 5.2 Sources and variability of $CO_2$ at 5 m depth

$CO_2$ concentrations at 5 m depth vary considerably over the sampling period and between individual sites (Fig. 5a). $CO_2$ concentrations are generally higher in the forest boreholes (Deep 1, Deep 2) than in the meadow boreholes (Deep 3, Deep 4), likely due to the high volumes of $CO_2$ produced by autotrophic respiration of the mature woodland root system which penetrates deep into the epikarst. The significant negative correlation between $pCO_2$ and MMT is highly influenced by borehole





Deep 1, where very high $CO_2$ concentrations were measured in June 2022 and June 2023 (Fig. 5b). The $CO_2$ $\delta^{13}C$ values across

all boreholes reflects the isotopic value of the C3 dominated ecosystem of the catchment, and does not indicate any C4 plant input from the nearby fields. Notably, the ecosystem composition of the overlying vegetation (meadow vs. forest) influences the isotopic composition of $CO_2$ at 5 m in the same way as the shallow boreholes, with a slightly lower $\delta^{13}C$ in the meadow than the forest. The $F^{14}C$ values of epikarst $CO_2$ vary across all deep boreholes by ± 0.02 around a mean value of 1.00 (Fig. 5c) suggesting that the $F^{14}C$ composition of $CO_2$ occurring in the shallower soil zone is from modern tree root $CO_2$ production

(Breecker et al., 2012; Tune et al., 2020). Furthermore, the year-round modern $CO_2$ $F^{14}C$ values in the deeper meadow boreholes Deep 3 and Deep 4 compared to the seasonally pronounced signal in the shallower boreholes of Shallow 2, highlights the potential influence of the meadow doline morphology. Boreholes Deep 3 and Deep 4 were drilled on the edge of the doline beneath shallow leptosols like those found in the forest, while Shallow 2 was drilled fully within the doline. Unfortunately, there is no epikarst depth borehole installed within the lower basin of the meadow to verify whether the seasonal signal

measured at Shallow 2 translates to the deeper subsurface.

Whilst many studies point to the degradation of exported aged organic matter being the main source of karstic ground air (Breecker et al., 2012; Mattey et al., 2016; Bergel et al., 2017), epikarst $CO_2$ in the Milandre cave catchment has predominantly modern $F^{14}C$ values, and does not show seasonal isotopic variability. This suggests that the majority of the $CO_2$ at 5 m is likely

supplied by contemporaneous C3 tree root respiration. This also excludes the contribution of substantial amounts of decadal-aged soil material containing bomb spike carbon due the enrichment in $^{14}C$ by thermonuclear weapons testing during 1950's and 1960's (Trumbore, 2000; Shi et al 2020). This result supports the body of literature stating that, in addition to the export and respiration of older carbon, large volumes of modern carbon are also transported into the unsaturated zone, likely in large part through deep root respiration (Breecker et al., 2012; Campeau et al., 2019; Tune et al., 2020). This suggests that deep

roots, as found in mature forest ecosystems, might be more important than previously thought in contributing to the ground air budget.

### 5.3 Ventilation driven isotopic variation in cave air

The $CO_2$ dynamics in the cave show distinct variations in concentrations and isotopic compositions between the two sampling points Upstream and Downstream (Fig. 6a, b, c). Cave ventilation dynamics modulate the mixing ratio between the atmosphere

and the other contributing pools, and are usually the dominant source of variability (Kukuljan et al., 2021; Buzjak et al., 2024). Recent modelling of air flow dynamics in Milandre cave found that outside temperature controls 95 % of flow variability (Garagnon et al., 2022). During colder months when the outside temperature is ≤ 8 ˚ C, the air in the cave flows topographically upwards from the northern downstream entrance close to the Downstream site, to the higher entrance closer to Upstream. When the temperature increases above 8 ˚C, the ventilation regime reverses. This seasonality is reflected in the alternating

concentrations and isotopic composition of the $CO_2$ at the cave sampling sites as they experience varying amounts of dilution from atmospheric air during the year. $CO_2$ degassing of the cave river at Upstream reduces the dilution effect in the upper





passage. The effects of ventilation can also be observed in the higher spatial resolution sampling during the cross trip (Fig. 8a, b, c) whereby the summer regime ventilation resulted in lower concentrations, more positive $\delta^{13}C$, and increased $F^{14}C$ due to atmospheric mixing closer to the upper entrance (Upstream site). Spatially, a marked decrease in $CO_2$ concentration and $F^{14}C$

occurs at sampling location 10 and 11 (~750 to 1000 m) from the cave entrance respectively, though little change is observed in the $\delta^{13}C$. This air input could be associated with an older organic matter pool which, however, contributes only little to the entire $CO_2$ mass flux.

## 5.4 Sources and variability of $CO_2$ in cave air

The Keeling plot y-intercepts show that the composition of the gas pool (karst endmember) that mixes with atmospheric air in the cave changes seasonally (Fig. 7c). It is important to note that, while the isotopic composition of cave $CO_2$ is influenced by ventilation-driven dynamics, the Keeling plot y-intercept represents the composition of the subsurface gas that mixes with the atmospheric air during ventilation and is not affected by it. The $\delta^{13}C$ of the karst endmember is mostly stable, varying -26 ‰, ± 2 ‰, a value typically associated with the $CO_2$ produced by the C3 plants which dominate the catchment ecosystem. This

value is also very similar to the $\delta^{13}C$ $CO_2$ values of soil and epikarst gas and reinforces the notion of a common source. Previous work investigating aquifer dynamics using $^{222}Rn$ at Milandre suggested that the majority of the cave $CO_2$ comes from the overlying soils (Savoy et al., 2011). The $F^{14}C$ on the other hand shows seasonal variability with distinctly more "modern" (closer to $F^{14}C = 1$) during the summer and "aged" ($F^{14}C < 1$) $CO_2$ during the winter. The transition from an $F^{14}C$ of 0.94 to 0.88 represents a change in the mean apparent age of $CO_2$ from ~ 100 to 1000 yrs, well beyond the assumed residence time of

water in the epikarst of ~ 5.5 to 6.6 years (Affolter et al., 2020).

Several mechanisms may be responsible for the seasonal $F^{14}C$ fluctuations of the endmember. Varying contributions of host rock-derived carbon to cave air through shifts in the host rock dissolution regime could affect the isotopic composition of cave air. Dissolution of the host rock carbonate can occur under a wide range of conditions between two extreme cases: in a

completely closed system, the aqueous solution becomes isolated from the soil $CO_2$ reservoir after passing into the epikarst resulting in a theoretical 50:50 contribution ratio of carbon ions in solution from the carbonate host rock and from soil air, and overall decreasing $F^{14}C$ to as low as 0.5 (Fohlmeister et al., 2011; Milanolo & Gabrovšek, 2015). On the other hand, in a completely open system the aqueous solution can continuously exchange with an unlimited soil $CO_2$ reservoir resulting in most carbon atoms sourced from the soil $CO_2$ reservoir, and the rock contribution being minimal (Fohlmeister et al., 2011).

Most natural systems exist in an intermediate state between fully open and fully closed settings.

If the seasonal variability in $CO_2$ $F^{14}C$ in Milandre cave reflected seasonal shifts in the "openness" of the system, summers (with higher $F^{14}C$) would be characterised by a more open system with higher rates of exchange between the aqueous solution and the soil $CO_2$ reservoir. The observed shift to lower $F^{14}C$ in winter would reflect a more closed system with a higher

contribution of the $F^{14}C$ free carbonate bedrock. Shifts in the open-closed continuum would likely result in changes in the $\delta^{13}C$



of the endmember (Fairchild & Baker, 2012) which is not seen in the cave air, with an expected lower $\delta^{13}$C in a more open system, and an increase in a more closed system because of the higher solid host rock carbon contribution ($\delta^{13}$C $\approx$ 0 ‰). However, varying amounts of isotopic fractionation can occur when $CO_2$ is degassed from the drip water DIC pool into the cave atmosphere, potentially influencing the isotopic composition of the measured cave air $CO_2$ (Mickler et al., 2019).


We constructed a mixing model which allows us to evaluate how much variability in the Keeling plot y-intercept (i.e. the second source of carbon to cave air besides the atmosphere) can be explained by variation in DIC contribution from dissolution regime changes and degassing fractionation. This model assumes that the Keeling plot y-intercept is itself a mixture of two endmembers that contribute $CO_2$ to cave air, the modern soil ($F^{14}C_{soil} = 1$), and the DIC ($F^{14}C_{DIC}$), where the $F^{14}C_{DIC}$ itself

represents a mixture of dissolved soil $CO_2$ and DIC from carbonate mineral dissolution that is degassed into the cave air. The $F^{14}C$ soil represents direct input of $CO_2$ soil gas into the cave.

The mixing ratio between $F^{14}C_{soil}$ and $F^{14}C_{DIC}$ results in the $F^{14}C$ of $CO_2$ entering the cave atmosphere ($F^{14}C_{cave}$). Thus, the cave air $CO_2$ $F^{14}C$ can be expressed as (where $f$ is fraction, defined as between 0 and 1):


$$F^{14}C_{cave} = f_{soil}F^{14}C_{soil} + f_{DIC}F^{14}C_{DIC} \qquad (3)$$

Solve for $f_{soil}$ and $f_{DIC}$, where $f_{DIC} = 1 - f_{soil}$:


$$F^{14}C_{cave} = f_{soil}F^{14}C_{soil} + (1 - f_{soil})F^{14}C_{DIC} \qquad (4)$$

For $F^{14}C$, fractionation is generally negligible during carbonate dissolution and degassing compared to $\delta^{13}$C, due to measurement precision i.e. % vs ‰ (Fohlmeister et al., 2011). We use the mixing calculation (Eq. 4) to estimate the contributions of $f_{soil}$ and $f_{DIC}$ using the highest (0.94), an intermediate (0.90), and the lowest (0.88) measured cave $CO_2$ $F^{14}C$

values ($F^{14}C_{cave}$), and $F^{14}C$ DIC ($F^{14}C_{DIC}$) based on the highest (0.98) intermediate (0.88) and lowest (0.78) measured values of drip water DIC. We find that when $F^{14}C_{DIC} = 0.98$, the mixing model results in impossible scenarios with $f_{soil} > 1$ and $f_{DIC}$ < 0 because this value is higher than the highest $F^{14}C_{cave}$, and a highly unlikely scenario based on our observations and the literature where $f_{soil} = 0$ and $f_{DIC} = 1$ when $F^{14}C_{DIC} = 0.88$, as this corresponds to the lowest $F^{14}C_{cave}$ value (Fig. 9a). Only the scenario with lowest $F^{14}C_{DIC} = 0.78$ produced realistic mixing fractions. Overall, this implies that there is a discrepancy

between the measured $F^{14}C_{DIC}$ and what is derived by a simplified mixing model based on the $F^{14}C$ of cave $CO_2$, with $F^{14}C_{DIC}$ values too high to result in the $F^{14}C_{cave}$ composition that we observe. Thus, carbonate dissolution and subsequent degassing of $CO_2$ from drip water DIC to cave air is unlikely to be the dominant process explaining the shifts in cave air $F^{14}C$ over time.



The $F^{14}C_{DIC}$ is based on the analysis of 9 drip water sites of varying drip rates and ecosystem coverage. It is possible that the
sampling set up did not capture the true range of $F^{14}C_{DIC}$ in the drip water, though rather unlikely due to the diversity in drip
rate and hydrological response of drips measured.

As our mixing ratios derived from measured DIC values are mostly not possible according to the two endmember mixing
scenario, we then explored how a wider theoretical range of $F^{14}C_{DIC}$ values (from closed to mainly open system dissolution
conditions, 0.50 to 0.90 F$^{14}$C) would affect the contributing fractions of $f_{soil}$ and $f_{DIC}$. Variations in $F^{14}C_{DIC}$: 0.50 to 0.85
result in viable mixing fractions across the range of $F^{14}C_{cave}$, with the $f_{soil}$ (20 to 76 %) and $f_{DIC}$ (24 to 80 %) differing
widely (Fig. 9b).

As the average value of cave CO$_2$ δ$^{13}$C is -26 ‰ (we take a range from -25 to -27 ‰), we created a similar two-endmember
mixing calculation for δ$^{13}$C values to test whether the mixing ratios derived from F$^{14}$C translate to observed δ$^{13}$C values. For
this we used the average δ$^{13}$C of soil CO$_2$ ($\delta^{13}C_{soil}$) = -26 ‰, and the δ$^{13}$C of the DIC ($\delta^{13}C_{DIC}$) = -15 ‰ (estimated from an
average of the δ$^{13}$C produced during the F$^{14}$C DIC AMS measurement (see Appendix B)).

Fractionation effects during degassing must be considered when modelling δ$^{13}$C. If equilibrium fractionation between DIC and
CO$_{2(g)}$ occurs, the fractionation factor ($\Delta^{13}C_{HCO3-CO2,eq}$) at 10.5 ˚C between HCO$_3$- and CO$_2$ is 9.54 ‰, with the gas being
9.54 ‰ depleted in $^{13}$C relative to the fluid (Mook et al., 1974). Kinetic fractionation can occur during rapid degassing of CO$_2$
as well as during rapid precipitation of carbonate minerals (Mickler et al., 2019). If kinetic fractionation occurs during
degassing, this results in a greater depletion of the dissolved CO$_2$ in $^{13}$C compared to equilibrium fractionation (i.e. a more
positive $\delta^{13}C_{DIC}$). The possible ranges reported for the composition of CO$_2$ generated from kinetic fractionation during
degassing in caves vary widely, with some reporting $\Delta^{13}C_{HCO3-CO2,kin}$ close to that resulting from equilibrium fractionation
(Dulinski & Rozanski et al., 1990), and others a range of 10 to 65 ‰ (Mickler et al., 2019). The extent of kinetic fractionation
is expected to depend on the amount of degassing that occurs, which is a function of the disequilibrium between the fluid and
the surrounding air (Frisia et al., 2011). In our system with limited geochemical data regarding fluid compositions, it is difficult
to determine what extent of kinetic fractionation may have occurred. Hence, we first assume that the fractionation is < -9.54
‰ (i.e., greater compared to equilibrium conditions) and take a moderate but arbitrary value ballpark of $\Delta^{13}C_{HCO3-CO2,kin}$ =
30 ‰.

A mixing calculation is then applied to assess if the measured range of the karst endmember δ$^{13}$CO$_2$ given by the Keeling plot
can be consistent with the extended range of $F^{14}C_{DIC}$ and measured $F^{14}C_{cave}$ under conditions with equilibrium degassing and
kinetic degassing:



Equilibrium degassing:

$$\delta^{13}C_{cave} = f_{soil}\delta^{13}C_{soil} + f_{DIC}(\delta^{13}C_{DIC} - \Delta^{13}C_{HCO3-CO2,eq}) \qquad (5)$$

Kinetic degassing:

$$\delta^{13}C_{cave} = f_{soil}\delta^{13}C_{soil} + f_{DIC}(\delta^{13}C_{DIC} - \Delta^{13}C_{HCO3-CO2,kin}) \qquad (6)$$

For the maximum (0.94), intermediate (0.90), and minimum (0.88) values of the endmember F[14]C, only equilibrium fractionation during degassing can explain the observed $\delta^{13}C_{cave}$ value of approximately -26 ‰ (Fig. 9c). Kinetic fractionation

scenarios result in very negative $\delta^{13}C_{cave}$ values which do not fit those measured, except when using very small kinetic fractionation factors similar to that of the equilibrium fractionation ($\Delta^{13}C_{HCO3-CO2,kin}$ of -10 ‰, see Appendix C). It is possible that kinetic fractionation occurs between Milandre cave waters and the cave air due to atmospheric ventilation which occurs year-round. The fresh atmospheric air which enters from the upper entrance in the summer and the lower entrance in the winter creates a concentration gradient between the $CO_2$ present within cave waters and the concentration in the cave atmosphere.

However, the rapidly increasing $pCO_2$ with distance from the cave entrance (Fig. 8a) implies that the concentration gradients required for kinetic fractionation do not occur throughout most of the cave passage and that kinetic influences are probably minimal.

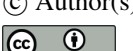



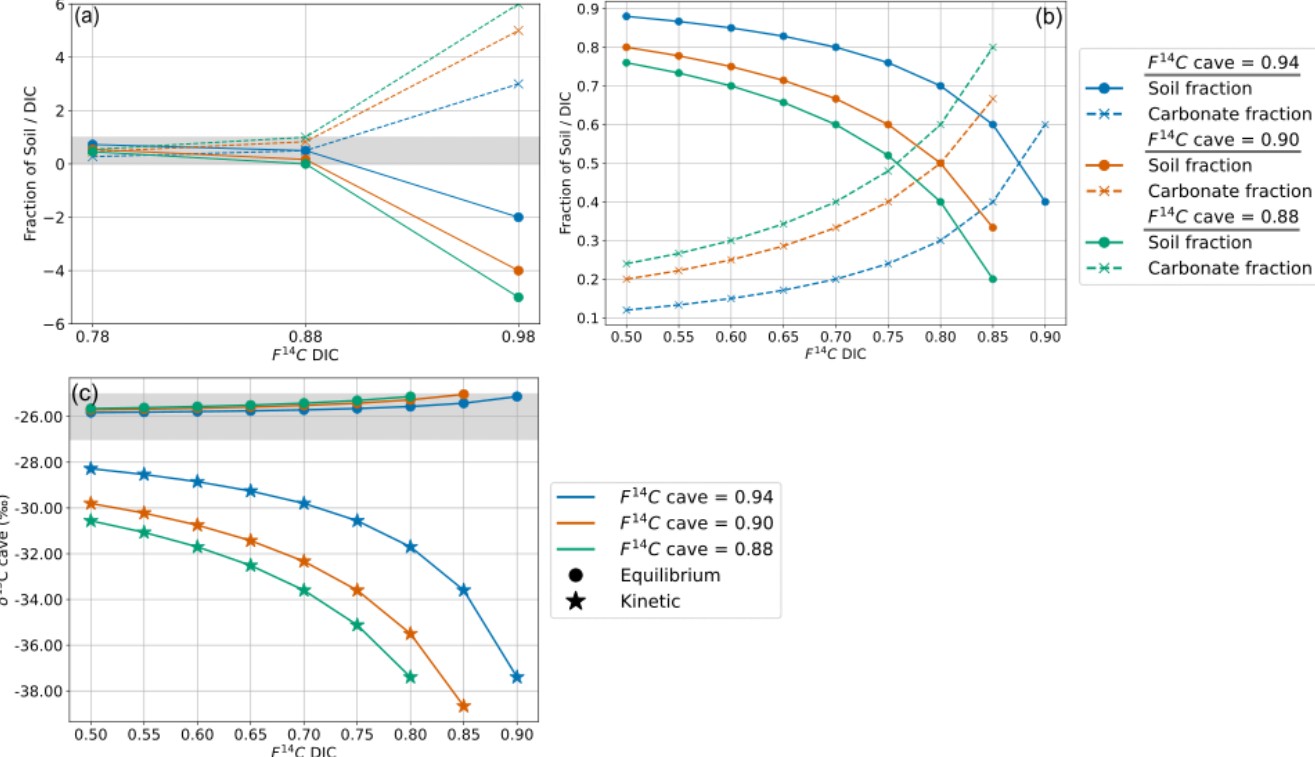

**Figure 9. a) Mixing model between $f_{soil}$ and $f_{DIC}$ for the measured range of F$^{14}$C DIC values and F$^{14}$C cave values of 0.94 (blue), 0.9 (orange) 0.88 (green). The grey bar shows physically possible mixing ratios between 0 and 1. b) Mixing model between $f_{soil}$ and $f_{DIC}$ using the extended DIC F$^{14}$C range and cave air F$^{14}$C values. c) Cave air δ$^{13}$C for varying DIC F$^{14}$C and cave air F$^{14}$C values with equilibrium fractionation at -9.54 ‰ (circles) and kinetic fractionation at -30 ‰ (stars). The grey bar shows cave air δ$^{13}$C range which fits the measured values between -25 ‰ and -27 ‰.**

Overall, the $F^{14}C_{DIC}$ measured beyond F$^{14}$C = 0.90 is too high to reproduce our measured cave air CO$_2$ F$^{14}$C and δ$^{13}$C. We measured the $F^{14}C_{DIC}$ in drips covering diverse locations, drip rates, and coverage so it is unlikely that our DIC data is not representative of the system. The $F^{14}C_{DIC}$ of the cave river was not measured, however previous studies on the Milandre cave river have suggested that in non-flooding conditions, the river is fed by slow diffuse flow as the epikarst acts as a buffer, storing water and dampening the signal of normal rainfall. During flooding events, the epikarst aquifer is bypassed and the stream is mainly fed by fresh fracture flow (Perrin et al., 2003; Savoy et al., 2011). A diffuse flow fed river is likely to have isotopic characteristics similar to that of the drip waters, which have higher F$^{14}$C suggesting more open system dissolution conditions. In contrast, times of flooding and soil saturation results in less exchange with the soil gas reservoir, leading to dissolution under more closed conditions (Perrin et al., 2003; Savoy et al., 2011). However, the drip water $F^{14}C_{DIC}$ is relatively stable over one year of monitoring in a variety of hydrological conditions, implying that variations in host rock dissolution and degassing dynamics are not the explanation for the variation in F$^{14}$C and δ$^{13}$C of the endmember.





The meadow soil boreholes in the doline (Shallow 2) have similar patterns in $CO_2$ composition to the endmember, with a $\delta^{13}C$ of -26 ‰ and variable $F^{14}C$ (Fig. 10a & b). In 2022, the $F^{14}C$ in the meadow soil and karst endmember align closely, but this decouples slightly in 2023 during the transitional periods between winter and spring where the karst endmember $F^{14}C$ changes before the meadow. This suggests that, instead of the soils influencing the cave air due to seasonal changes in soil respiration, the cave ventilation regime could be influencing the soil $F^{14}C$. The cave ventilates upwards in winter from Downstream to Upstream in a chimney effect (and vice versa in summer). Here, the lower $F^{14}C$ karst endmember $CO_2$ may flow upwards through the highly fractured epikarst of the doline and ventilate the meadow soils, resulting in a mixture of low $F^{14}C$ $CO_2$ from the karst endmember and modern $CO_2$ from the soils. This possibly reflects a contribution of older $CO_2$ from within the karst itself, likely from a "ground air" reservoir of older respiring organic material in the epikarst as found by several other studies (Breecker et al., 2012; Noroha et al., 2015; Mattey et al., 2016; Bergel et al., 2017). There are likely several reservoirs of older organic material throughout the downstream catchment of Milandre which may contribute the low $F^{14}C$ $CO_2$ to the ventilating cave air in winter. In summer the ventilation reverses, and the higher $F^{14}C$ $CO_2$ from the soil is transported downwards into the cave, increasing the $F^{14}C$ of the cave endmember $CO_2$. The $F^{14}C$ of the karst endmember is always lower than the meadow soil $F^{14}C$, implying that there is always mixing between the soils and the karst endmember. Similar upwards transport of $CO_2$ in karst boreholes has been observed in Nerja Cave, Spain (Benavente et al., 2010; Benavente et al., 2015), and in the Gibraltar karst (Mattey et al., 2016). We do not observe significant effects of the winter upwards ventilation in the $F^{14}C$ $CO_2$ in any of the deeper boreholes in the meadow or any of the boreholes in the forest. This could be related to the location of Shallow 2 in the doline, whilst the rest of the boreholes are situated higher on the banks (Deep 3 and Deep 4), or further away from the doline in the forest (Deep 1 and Deep 2). The topography likely reflects influences of the secondary porosity in the bedrock, which is possibly more highly fractured in the meadow doline formation compared to the forest leading to a greater effect of ventilation in the meadow area and an undisturbed modern $^{14}C$ signature of forest soils. As the doline structure likely has a higher secondary porosity and is more highly fractured than the higher bedrock, ventilation effects may be more important here than at the other locations.



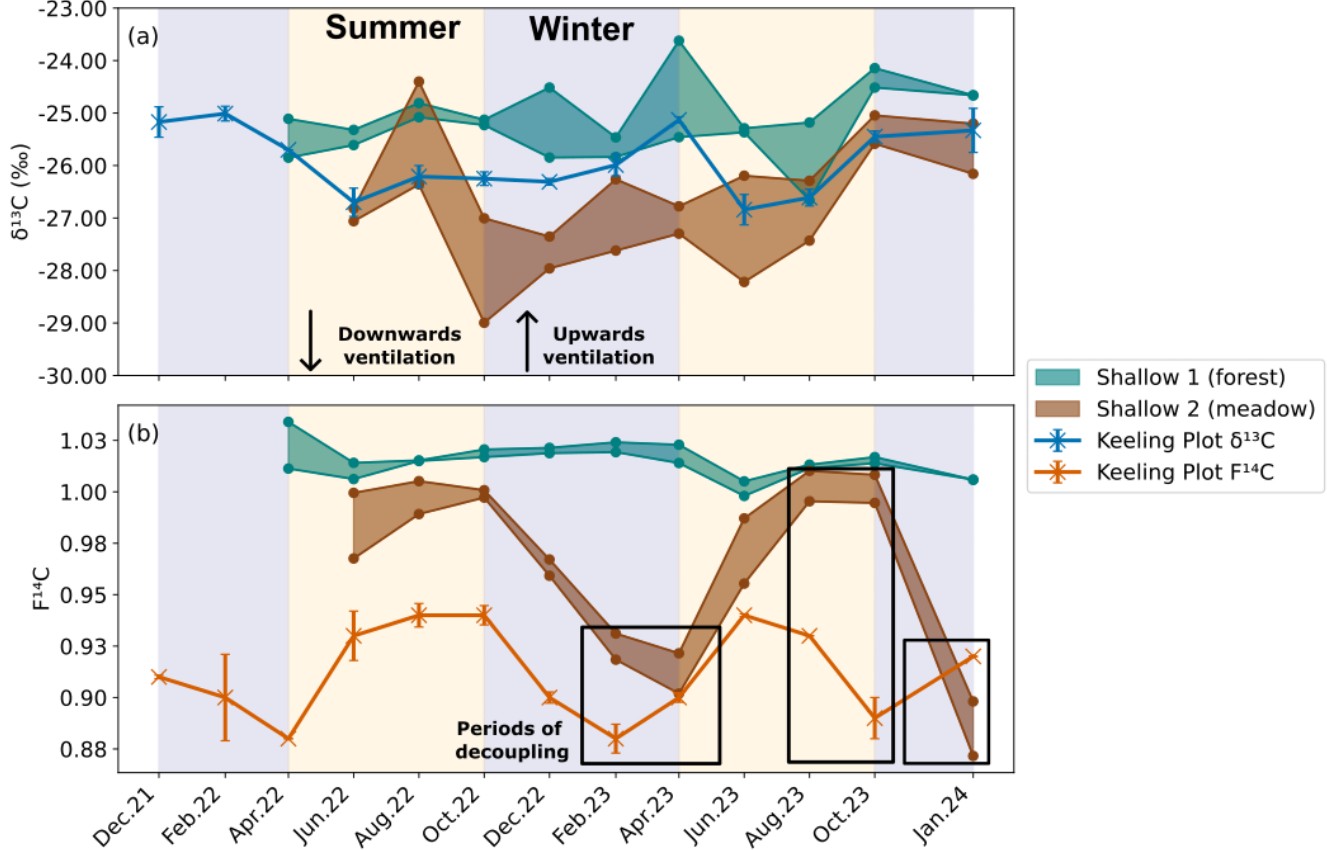

**Figure 10. Comparison between the a) δ¹³C and b) F¹⁴C of the shallow depth boreholes in the forest (Shallow 1, green) and meadow (Shallow 2, brown), and the Keeling plot endmembers δ¹³C (blue) and F¹⁴C (orange) also show in Figure 7. Summer (yellow) and winter (blue) seasons are highlighted by the coloured background. The direction of cave ventilation is shown by the arrows with downwards ventilation (from Upstream to Downstream) in summer, and upwards ventilation (Downstream to Upstream). Periods of decoupling between the F¹⁴C of the meadow borehole and the cave endmember are shown in black boxes.**

The accumulation of the older organic carbon in the epikarst likely occurs over time. Soil layers in karst regions often accumulate on bedrock with deep vertical fractures (Cheng et al., 2023). This decreases the ability of long-term storage of older soil carbon pools due to preferential fracture flow, also known as subsurface leakage, which destabilises the oldest soil layer and leads to export of older carbon into the epikarst (Sánchez-Cañete et al., 2018; Wan et al., 2018). Several studies have acknowledged this potential source of $CO_2$ produced autochthonously inside the epikarst by the microbial degradation of this old soil, plant, or root material which was previously washed into the unsaturated zone, contributing older $CO_2$ to the ground air mix (Noronha et al., 2015; Bergel et al., 2017; Ding et al., 2023).





## 6 Conclusion

We conducted a comprehensive investigation over two years into the dynamics of $CO_2$ concentrations, $\delta^{13}C$ and $F^{14}C$ composition across different land covers and soil types within the Critical Zone at Milandre cave in northern Switzerland. Our analysis found distinct seasonal fluctuations in $CO_2$ levels in soil boreholes in both forest and meadow areas, with higher concentrations during summer compared to winter. Low $\delta^{13}C$ values of $\sim$ -26 ‰ indicate respiration of C3 plants which dominate the catchment area. The stable modern $F^{14}C$ of $\sim$ 1.00 in shallow forest boreholes indicated a year-round modern

$CO_2$ contribution from tree and plant roots to the subsurface, consistent with a well-ventilated soil. On the other hand, samples from meadow boreholes situated in a doline with a thick, well developed soil cover displayed seasonality in $F^{14}C$ from $\sim$ 0.88 in the winter and spring to $\sim$ 1.00 in summer.

In the epikarst we observed substantial variability in $CO_2$ concentrations across the catchment ($\sim$ 3000 to $\sim$ 37'000 ppmV),

with higher concentrations in boreholes with forest cover than those with meadow cover. Despite differences in surface vegetation, the isotopic compositions of $\delta^{13}C$ and $F^{14}C$ remained stable in both forest and meadow environments, reflecting the dominant modern C3 vegetation signature contributing to the ground air. The year-round modern epikarst $CO_2$ in meadow boreholes Deep 3 and Deep 4, in contrast to the seasonally attenuated signal in the thicker meadow soils.

Large variations in cave air $CO_2$ concentrations (atmospheric to $\sim$ 30'000 ppmV), $\delta^{13}C$ ($\sim$ -9 to -25 ‰), and $F^{14}C$ (0.88 to 0.98), are controlled by direction changes in the seasonal temperature driven cave ventilation regime. The seasonal changes in the isotopic composition of the Keeling plot derived karst endmember cannot be explained with the observed DIC $F^{14}C$ range, suggesting that bedrock dissolution and degassing dynamics are likely not the cause of the isotopic variation in the cave air. The isotopic characteristics of the cave air are comparable to those of the meadow doline soil. The karst endmember $CO_2$ $F^{14}C$

changes before the soil in times of variation, and is always older. This suggests that the cave ventilation contributes older $CO_2$ to the doline soils, likely sourced from a reservoir of aged organic material in the epikarst, during the upward winter ventilation regime. In summer, the reversed downwards ventilation contributes younger soil $CO_2$ to the cave, mixing with the older epikarst reservoir. The consistently low $\delta^{13}C$ signature of the karst endmember implies that the impact of carbonate dissolution on the system is low, and that the $CO_2$ is contributed from either modern soils or aged organic material in the epikarst.


Our study provides important insights into how carbon is transported to the subsurface in karstic Critical Zones. Understanding these processes is crucial for accurate estimation of the size of subsurface $CO_2$ pools (ground air), and to refine terrestrial $CO_2$ budgets. Moving forward, it would be beneficial for future research to undertake more detailed investigation of $CO_2$ transport into the subsurface implementing $^{14}C$ analysis to further constrain the sources of ground air. Furthermore, higher resolution

monitoring over periods of interest, could assist us in understanding short-term variations.



**Appendix A**



**A1 Spearman's rank correlation matrix displaying the Spearman's rho of CO₂ concentration, δ¹³C and F¹⁴C, Mean Monthly Temperature (MMT) and Mean Monthly Precipitation (MMP) from Fahy weather station from samples from soil depth boreholes Shallow 1. Parameters with a significant correlation (positive or negative) are bolded and the strength of the correlation is indicated by the blue – red colour bar.**




**A2 Spearman's rank correlation matrix displaying the Spearman's rho of CO₂ concentration, δ¹³C and F¹⁴C, Mean Monthly Temperature (MMT) and Mean Monthly Precipitation (MMP) from Fahy weather station from samples from soil depth borehole Shallow 2. Parameters with a significant correlation (positive or negative) are bolded and the strength of the correlation is indicated by the blue – red colour bar.**

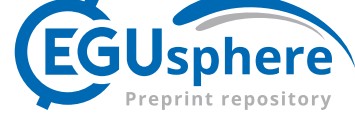



**A3 Spearman's rank correlation matrix displaying the Spearman's rho of CO₂ concentration, δ¹³C and F¹⁴C, Mean Monthly Temperature (MMT) and Mean Monthly Precipitation (MMP) from Fahy weather station from samples from epikarst depth boreholes Deep 1 and Deep 2. Parameters with a significant correlation (positive or negative) are bolded and the strength of the correlation is indicated by the blue – red colour bar.**




A4 Spearman's rank correlation matrix displaying the Spearman's rho of $CO_2$ concentration, $\delta^{13}C$ and $F^{14}C$, Mean Monthly
Temperature (MMT) and Mean Monthly Precipitation (MMP) from Fahy weather station from samples from epikarst depth
boreholes Deep 3 and Deep 4. Parameters with a significant correlation (positive or negative) are bolded and the strength of the
correlation is indicated by the blue – red colour bar.







**A5 Spearman's rank correlation matrix displaying the Spearman's rho of CO₂ concentration, δ¹³C and F¹⁴C, Mean Monthly Temperature (MMT) and Mean Monthly Precipitation (MMP) from Fahy weather station from samples from the Downstream cave site. Parameters with a significant correlation (positive or negative) are bolded and the strength of the correlation is indicated by the blue – red colour bar**



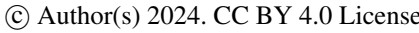

**A6 Spearman's rank correlation matrix displaying the Spearman's rho of $CO_2$ concentration, $\delta^{13}C$ and $F^{14}C$, Mean Monthly Temperature (MMT) and Mean Monthly Precipitation (MMP) from Fahy weather station from samples from the Upstream cave stie. Parameters with a significant correlation (positive or negative) are bolded and the strength of the correlation is indicated by the blue – red colour bar.**






**B1 Drip water dissolved inorganic carbon F$^{14}$C and AMS derived δ$^{13}$C from the actively dripping galleries nearby Upstream and Downstream.**

| Drip | Sampling date | F$^{14}$C | F$^{14}$C % error | AMS δ$^{13}$C (‰) |
|---|---|---|---|---|
| GF1 | Dec.21 | 0.86 | 0.92 | -13.6 |
| GF2 | Dec.21 | 0.94 | 0.91 | -14.3 |
| GR2 | Dec.21 | 0.91 | 0.92 | -12.4 |
| GR3 | Dec.21 | 0.98 | 0.93 | -13.5 |
| GR4 | Dec.21 | 0.91 | 0.97 | -14.7 |
| GR5 | Dec.21 | 0.90 | 0.97 | -14.2 |
| GR6 | Dec.21 | 0.85 | 1.02 | -13.6 |
| GF1 | Feb.22 | 0.87 | 0.99 | -12.6 |
| GF3 | Feb.22 | 0.95 | 0.96 | -12.7 |
| GR3 | Feb.22 | 0.90 | 0.97 | -14.1 |
| GR4 | Feb.22 | 0.89 | 0.99 | -12.7 |
| GR6 | Feb.22 | 0.84 | 1.00 | -11.9 |
| GR6 | Feb.22 | 0.88 | 0.99 | -10.6 |
| GF1 | Apr.22 | 0.88 | 0.93 | -12.8 |
| GF2 | Apr.22 | 0.93 | 0.89 | -11.6 |
| GF3 | Apr.22 | 0.98 | 0.86 | -12.1 |
| GR1 | Apr.22 | 0.87 | 0.90 | -12.6 |
| GR2 | Apr.22 | 0.89 | 0.92 | -12.1 |
| GR3 | Apr.22 | 0.90 | 0.93 | -12.2 |
| GR4 | Apr.22 | 0.90 | 0.93 | -13.4 |
| GR5 | Apr.22 | 0.89 | 0.98 | -13.7 |
| GR6 | Apr.22 | 0.90 | 0.99 | -13.6 |
| GF1 | Jun.22 | 0.87 | 1.11 | -18.0 |
| GF2 | Jun.22 | 0.93 | 1.11 | -18.9 |
| GF3 | Jun.22 | 0.96 | 1.07 | -18.6 |
| GR1 | Jun.22 | 0.88 | 1.12 | -17.4 |
| GR2 | Jun.22 | 0.93 | 1.09 | -17.6 |
| GR3 | Jun.22 | 0.89 | 1.12 | -17.9 |
| GR4 | Jun.22 | 0.89 | 1.12 | -18.9 |
| GR5 | Jun.22 | 0.90 | 1.11 | -17.9 |
| GR6 | Jun.22 | 0.92 | 1.12 | -19.0 |
| GF1 | Aug.22 | 0.86 | 1.10 | -17.1 |
| GF2 | Aug.22 | 0.92 | 1.12 | -19.8 |
| GR1 | Aug.22 | 0.87 | 1.03 | -8.9 |



| GR2 | Aug.22 | 0.91 | 1.09 | -18.3 |
| GR4 | Aug.22 | 0.93 | 1.10 | -19.2 |
| GR5 | Aug.22 | 0.91 | 1.10 | -19.5 |
| GR6 | Aug.22 | 0.93 | 1.11 | -20.8 |
| GF1 | Oct.22 | 0.84 | 1.05 | -9.4 |
| GF2 | Oct.22 | 0.94 | 1.12 | -20.3 |
| GF3 | Oct.22 | 0.95 | 1.11 | -20.8 |
| GR1 | Oct.22 | 0.86 | 1.14 | -17.2 |
| GR2 | Oct.22 | 0.88 | 1.14 | -22.6 |
| GR4 | Oct.22 | 0.91 | 1.13 | -18.7 |
| GR5 | Oct.22 | 0.89 | 1.13 | -18.3 |
| GR6 | Oct.22 | 0.91 | 1.11 | -19.1 |
| GF1 | Dec.22 | 0.86 | 1.01 | -14.1 |
| GF2 | Dec.22 | 0.88 | 0.94 | -14.1 |
| GF3 | Dec.22 | 0.94 | 0.95 | -11.1 |
| GR1 | Dec.22 | 0.86 | 1.00 | -11.5 |
| GR4 | Dec.22 | 0.89 | 0.98 | -11.5 |
| GR5 | Dec.22 | 0.89 | 1.01 | -12.6 |
| GR6 | Dec.22 | 0.78 | 1.02 | -12.9 |





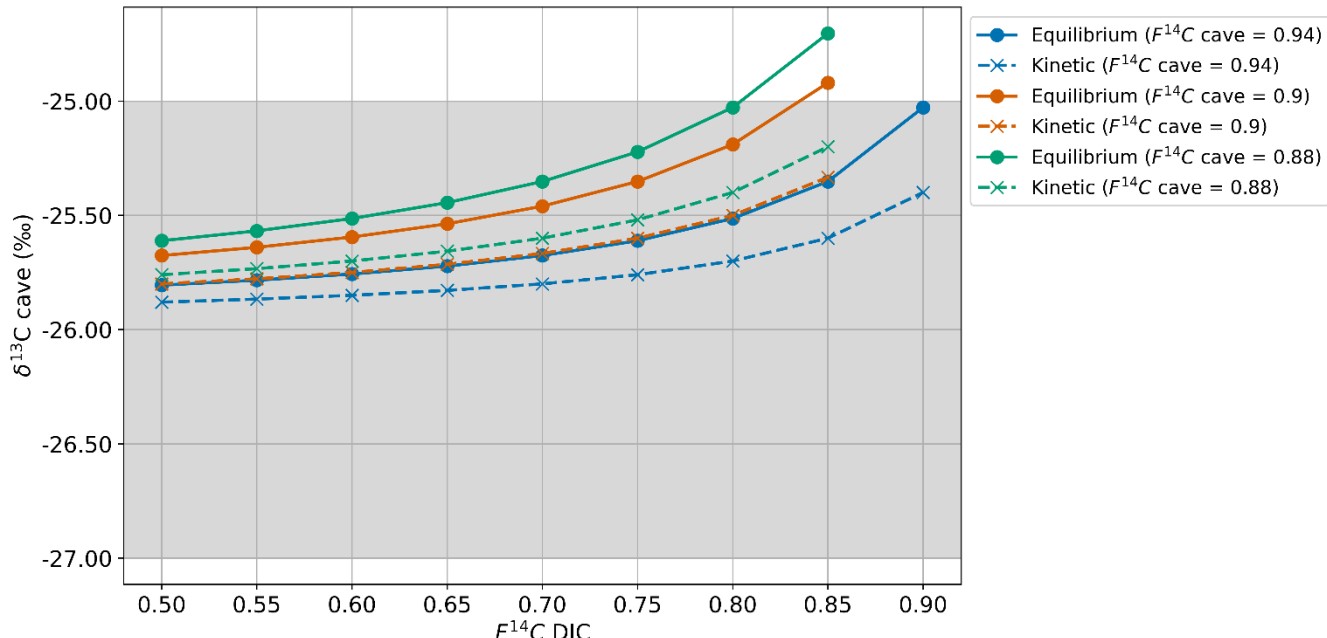

**C1 Cave air δ¹³C for varying DIC F¹⁴C and cave air F¹⁴C values with equilibrium fractionation at -9.54 ‰ (solid lines) and kinetic fractionation at -10 ‰ (dashed lines).**


**Code and data availability**

The data, code for Spearman's rank statistics, and code to reproduce the model results are publicly available on Zenodo at https://doi.org/10.5281/zenodo.14253707.

**Author contributions**

Sarah Rowan: Conceptualization, Formal analysis, Investigation, Methodology, Visualization, Software, Writing – original draft preparation.

Marc Luetscher: Conceptualization, Resources, Visualization, Writing – review & editing.

Thomas Laemmel: Methodology, Validation, Writing – review & editing.

Anna Harris: Methodology, Writing – review & editing.

Sönke Szidat: Writing – review & editing.

Franziska Lechleitner: Conceptualization, Funding acquisition, Project administration, Resources, Supervision, Writing – review & editing.



**Competing interests**

The authors declare that they have no conflict of interest.

**Disclaimer**

**Acknowledgements**

This work was supported by the Swiss National Science Foundation Ambizione grant number 186135. We gratefully thank the staff of the Swiss Institute of Speleology and Karst Studies for their previous and continued work in Milandre cave, and for their contribution of field assistance and expertise in karst science. Special thanks to Claudio Pastore for his assistance during the cross trip. We express our gratitude to the Spéléo Club Jura for the initial exploration of Milandre cave, and for their persistent work and maintenance of the area. We also appreciate Daniel Pape, for continued access to the cave entrance and boreholes. Thank you also to Madalina Jaggi, Dr. Oliver Kost, and Prof. Heather Stoll for providing access and support using the Picarro system. Graphitizations and AMS measurements were also assisted by Michael Staub and Dr. Gary Salazar. Generative AI (ChatGPT versions 3.5, 4 and 4o) was used for code troubleshooting.

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
