# Peer review of "Subsurface CO2 dynamics in a temperate karst system reveal complex seasonal and spatial variations"

_EGUsphere, 2024_

## Author Comment (AC1)

Response to Reviewer 1

We greatly appreciate the detailed review of our article, and thank the reviewer for their insightful suggestions and edits. We have taken the recommendations onboard and will edit the manuscript according to the following responses.

**General Comments**

The present study adds to the knowledge of CO2 cycling in the critical zone. It uses Grotte de Milandre as case study to investigate the unsaturated zone. This is attained by an extensive monitoring of the pCO2, stable C isotope ratios and 14C02 from shallow soil (under meadows and forest), to epikarst to cave atmosphere upstream (topographically high) and downstream (topographically low).

There is a wealth of information in the present research manuscript that is commendable. Yet, there are aspects of the systems that are not fully explored, which yield, in the end, to a conclusion that the deep root zone respiration is more important than we think.

I am no expert in rhizosphere, and in microbial processes. I am just a humble mineralogist. But, by exploring how crystals grow in both tropical and polar environments, I am aware that sources of CO2 are multiple and complex. It is well known to those who work on Antarctic subglacial environments that microbes thrive whenever there is water. Temperature is not the critical factor, but the absence of liquid H2O...on Earth.

In the dark cave environment, microbes and fungi thrive, and their respiration must influence the cave air pCO2. Unique chemoautotrophic communities have been described for Kartchner cave, with potentially multiple CO2 fixation pathways (Ortiz et al., 2014). Cave methanotrophic and heterotrophic bacteria are producers of bioactive compounds acting as rapid CO2 sources and sinks (Martin-Pozas et al., 2020). Cave sediments appear to play an active role in carbon cycle (Martin-Pozas et al., 2022). And a role for cave sediments is not really discussed in the present research paper. Yet, I do not know if there are sediments in Grotte de Milandre, as I could not find any references related to that specific topic. Perhaps there are no sediments. So, this should be stated somewhere in the text.

Various types of clastic cave sediments are indeed present in Milandre cave, including pebbles, gravels and clay deposits. The microbial activity in these deposits has not yet been investigated and was out of scope of the present study. Although associated life is undoubtedly present, microbial activity and thus $CO_2$/$CH_4$ is likely to be fostered by episodic anthropogenic pollution of the cave river. This point will be specifically addressed in the revised paper, stressing out the need for further investigations. The strong seasonality observed both in the soil $pCO_2$ and cave

pCO$_2$, however, advocates in favour of a circadian rhythm which, in our understanding, is difficult to accommodate with endogenic processes.

The importance of deep rhizosphere has been already pointed out in karst studies (Iversen, 2010; Chen, 2019; Wen et al., 2021). In contrast, the importance of the cave autochthonous microbial assemblages remains poorly known.

We thank the reviewer for these comments and we will certainly investigate these aspects more thoroughly in a future study. It is true that our manuscript in the present state does not put large specific emphasis on the respiration of cave microbial and fungal communities. While we were not able to assess the contributions to the unsaturated zone CO$_2$ budget from these potential sources, we will add these considerations to the discussion to highlight their theoretical relevance (see below). We will also add a few sentences describing the clay sediments to the site description of Milandre cave and stress out the link observed between suspended particles and microbial activity during flood events, in particular.

Line ~80: "Clay and silts, deposited during hydraulic backflooding, are particularly present in the downstream part of Milandre cave where they contribute to the sediment load during minor flood events. In contrast, during medium to intense flow, allochtonous sediment transport is often associated with higher microbial concentrations (Vuilleumier et al., 2021). Quantifying the contribution of microbial respiration to the cave carbon cycle was beyond the scope of this research and should be addressed by future research."

Vuilleumier, C., Jeannin, P.-Y., Hessenauer, M., & Perrochet, P. (2021). Hydraulics and turbidity generation in the Milandre Cave (Switzerland). *Water Resources Research*, 57, e2020WR029550. https://doi. org/10.1029/2020WR029550

In particular, cave sediments and speleothems (and soils above caves) may host methanotrophs, which may change the 13C values of both soil and cave atmosphere (Waring et al., 2017). As it has been demonstrated that caves host thriving microbial (including fungi) communities, the discussion should tackle this aspect of the cave dynamics. Not just ventilation, but in-situ microbial processes that occur in both oxidising and reducing "local" settings.

In the framework of understanding C cycling, it is important to advance our understanding of C release and sinks in the subterranean environment (Zheng et al., 2024). One of the C-cycle contributors in caves is methane. Thus, more of the discussion should be focused on ruling out methane...as this gas can also be evolved from ancient pore fluid waters trapped in the Oxfordian marls. Old methane can be ruled out by F 14C, but in-cave produced C from methane cannot.

Thank you for this comment. It is true that we cannot exclude a contribution to cave CO$_2$ from the oxidation of methane, either from methanotrophs or through the evolution from pore fluid waters.

Methane can be produced by both thermogenic and biogenic sources (Suess & Whiticar, 1988). The $\delta^{13}C$ $CO_2$ range of published values for the $CO_2$ resulting from methane oxidation from both sources is broad, with biogenic spanning from ~ - 20 to -90 ‰, and thermogenic from ~ -30 to - 65 ‰ (Suess & Whiticar, 1988). The $\delta^{13}C$ of $CO_2$ produced by thermogenic sources can also evolve to ~ - 25 to - 34 ‰ in settings where there is near total conversion of $CH_4$ to $CO_2$ (Shoell, 1980; Frieling et al., 2016).

While measuring methane concentrations quantitatively was not possible with our analytical setup, qualitative data shows that methane concentrations in Milandre cave air and catchment borehole samples were exceedingly low (~ 90 to 970 ppb), suggesting methane oxidation may occur in the Milandre subsurface environment.

Methanotrophic bacteria have been identified in 98 % of soils sampled from 21 North American caves at a mean bacterial abundance of 0.88 % (Webster et al., 2022).  Methanotrophic bacteria in caves significantly contribute to methane consumption, with a study in Pindal Cave (NW Spain) showing methanotrophic bacteria associated with moonmilk deposits remove 65 % to 90 % of atmospheric methane (Martin-Pozas et al., 2022).

The reviewer is correct that this could affect the isotopic ratios of cave $CO_2$. Isotopically, methanotrophs preferentially uptake the lighter $^{12}C$, resulting in a depleted $^{13}C$. Methane that undergoes biogenic oxidation is typically sourced from the atmosphere, meaning the resulting $CO_2$ will have a modern $F^{14}C$ signature ($F^{14}C$ ~ 1). In contrast, $CO_2$ produced by thermogenic methane oxidation will have a fossil $F^{14}C$ signature, as the methane originates from ancient organic rich sediments that contain no measurable radiocarbon ($F^{14}C$ ~ 0). As our cave and borehole $CO_2$ samples have a $\delta^{13}C$ of ~ -26 ‰ and a variable $F^{14}C$ in the cave air and shallow meadow samples, we cannot rule out a contribution of methane oxidation from biogenic processes actively occurring in Milandre, but this would not explain the depleted radiocarbon signature that we measured episodically in the cave air.

Methane evolved from pore fluids in Oxfordian marls may have an isotopic ratio similar to that measured in our samples if thermogenically derived with a dominant conversion of $CH_4$ to $CO_2$. However, it is unlikely that this process occurred in the Milandre cave sediments given the lack of organic rich sedimentary rocks in the stratigraphy.

Thus, while we cannot quantify the importance of methane oxidation on the subsurface carbon budget at our site, it is unlikely that methane oxidation processes are responsible for the variation in radiocarbon ages in cave air $CO_2$. We will add a paragraph explaining the importance of these processes in the subsurface to the discussion at Line ~ 470.

Frieling, J., Svensen, H. H., Planke, S., Cramwinckel, M. J., Selnes, H., & Sluijs, A. (2016). Thermogenic methane release as a cause for the long duration of the PETM. Proceedings of the National Academy of Sciences, 113(43), 12059–12064. https://doi.org/10.1073/pnas.1603348113

Martin-Pozas, T., Cuezva, S., Fernandez-Cortes, A., Cañaveras, J. C., Benavente, D., Jurado, V., Saiz-Jimenez, C., Janssens, I., Seijas, N., & Sanchez-Moral, S. (2022). Role of subterranean microbiota in the carbon cycle and greenhouse gas dynamics. Science of The Total Environment, 831, 154921. https://doi.org/10.1016/j.scitotenv.2022.154921

Schoell, M. (1980). The hydrogen and carbon isotopic composition of methane from natural gases of various origins. Geochimica et Cosmochimica Acta, 44(5), 649–661. https://doi.org/10.1016/0016-7037(80)90155-6

Suess, E., & Whiticar, M. J. (1989). Methane-derived CO2 in pore fluids expelled from the Oregon subduction zone. Palaeogeography, Palaeoclimatology, Palaeoecology, 71(1), 119–136. https://doi.org/10.1016/0031-0182(89)90033-3

Webster, K.D. *et al.* (no date) 'Diversity and Composition of Methanotroph Communities in Caves', *Microbiology Spectrum*, 10(4), pp. e01566-21. Available at: https://doi.org/10.1128/spectrum.01566-21.

An old isotope study on stable C and O isotope ratios in karst spring waters, where catchments were under forest and meadow soils in karst systems (Stichler et al., 1997) should be discussed to see how the Grotte de Milandre dataset compares with the 1997 one from Slovenia.

We thank the review for the reference to this extensive work in Slovenia. The Stichler et al.,1997 paper indeed compared the $\delta^{18}O$ in different karst springs, however the paper is lacking a description of the catchment vegetation and interpretation of the results. As we did not measure the $\delta^{18}O$ of groundwater in this study, it is difficult to discuss the findings from Stichler et al. in the context of our study. However, the next paper in the journal from Urbanc et al. looked at the $\delta^{13}C$ of soil $CO_2$ and spring water in different forested and meadowed sites. Figure 5.45 (page 254) is particularly interesting, where sites with more forest cover have a 1 permille higher soil $CO_2$ $\delta^{13}C$ than sites with meadow cover. This is similar to what we find in our boreholes (Fig.3 and Fig.5).

We would suggest adding a sentence to Line 375 comparing our results to the Urbanc study: "Though both meadow and forest boreholes have C3 plant 375 $\delta^{13}C$ signatures, the $\delta^{13}C$ of the meadow soils are on average lower than those in the forest soils. Previous analysis of soil $CO_2$ from various karstic sites in Slovenia showed a similar trend, whereby the $\delta^{13}C$ of meadow sites had a ~1 ‰ lower $\delta^{13}C$ compared to forested sites (Stickler et al., 1997)."

Stickler, W., Trimborn, P., Maloszewski, P., Rank, D., Papesch, W., Reichert, B. (1997). Environmental Isotope Investigations. *Acata Carsologica*, 26/1, 236 – 259.

Overall, this reviewer believes that more insight should be given to the complexity of the rhizosphere by reporting more results from soil research. It should also provide more insight in the discussion about the contribution of cave microbial communities. If (as I would seem) no research has been carried out on fingerprinting the cave microbial communities as it was done elsewhere (see for example Tomczyk-Żak and Zielenkiewicz, 2017; Kosznik-Kwa´snicka et al., 2022; Lange-Enyedi et al., 2022; Lange-Enyedi et al., 2023; Gogoleva et al., 2024) then the "in-cave" contribution...which would be associated with the "ventilation" process (as in-situ microbes are unaccounted for) would go...zilch.

We thank the reviewer for this comment. We will strengthen our discussion by summarising the diversity of microorganisms potentially present in caves such as Milandre, their potential to contribute $CO_2$ to the Critical Zone, and why we cannot distinguish these different organic sources as part of the discussion in sections 5.2 and 5.4 of the discussion.

Despite the apparent limiting factors for microbial growth in caves, it is known that different organisms are able to thrive within this niche including bacteria, archaea, algae, and fungi. In terms of bacteria, Proteobacteria is typically reported as the dominant group in caves (Zhou et al., 2007; Tomczyk-Żak and Zielenkiewicz, 2016). However, in some cases Actinobacteria is the largest group of microbes (Wiseschart et al., 2018). In two Chinese caves with high $CO_2$ concentrations (> 3000 ppm), Proteobacteria, Actinobacteria, Bacteroidetes, and *Thaumarchaeota* were the dominant phyla (Chen et al., 2023). Hundreds of fungi taxa have been described in caves globally, with Ascomycota being the most commonly reported (Gherman et al., 2014). Though not yet thoroughly investigated, diverse communities of protists have also been identified within caves (Gogoleva et al., 2024), as well as archaea (Biagioli et al., 2023), and algae (Kosznik- Kwaśnicka et al., 2022). As biological processes generally favour the incorporation of lighter $^{12}C$ compared to $^{13}C$, the respiration of these organisms would contribute an organic signature (lower $\delta^{13}C$ and a $F^{14}C$ corresponding to the age of the carbon source being consumed, which can range from modern to more aged $F^{14}C$) to the $CO_2$ composition of cave air at Milandre cave. With the isotopic analysis used in this study, and because we did not analyse the microbial community diversity present in Milandre cave and its catchment, the $CO_2$ contributions from different types of microbial processes cannot be distinguished from one another. We are only able to distinguish between "organic" processes (in-cave microbial, soil, root, and fungi respiration) which produce a $CO_2$ with a lower $\delta^{13}C$ signature and a range of $F^{14}C$ depending on the age of the organic material consumed, and inorganic processes (carbonate degassing and atmospheric ventilation) which contribute $CO_2$ with a higher $\delta^{13}C$ and more aged $F^{14}C$.

Biagioli, F., Coleine, C., Piano, E., Nicolosi, G., Poli, A., Prigione, V., Zanellati, A., Varese, C., Isaia, M., & Selbmann, L. (2023). Microbial diversity and proxy species for human impact in Italian karst caves. *Scientific Reports*, *13*(1), 689. https://doi.org/10.1038/s41598-022-26511-5

Chen, J., Li, Q., He, Q., Schröder, H. C., Lu, Z., & Yuan, D. (2023). Influence of CO2/HCO3– on Microbial Communities in Two Karst Caves with High CO2. *Journal of Earth Science*, *34*(1), 145–155. https://doi.org/10.1007/s12583-020-1368-9

Gherman, V. D., Boboescu, I. Z., Pap, B., Kondorosi, É., Gherman, G., & Maróti, G. (2014). An Acidophilic Bacterial-Archaeal-Fungal Ecosystem Linked to Formation of Ferruginous Crusts and Stalactites. *Geomicrobiology Journal*, *31*(5), 407–418. https://doi.org/10.1080/01490451.2013.836580

Gogoleva, N. E., Nasyrova, M. A., Balkin, A. S., Chervyatsova, O. Y., Kuzmina, L. Y., Shagimardanova, E. I., Gogolev, Y. V., & Plotnikov, A. O. (2024). Flourishing in Darkness: Protist Communities of Water Sites in Shulgan-Tash Cave (Southern Urals, Russia). *Diversity*, *16*(9), Article 9. https://doi.org/10.3390/d16090526

Kosznik-Kwaśnicka, K., Golec, P., Jaroszewicz, W., Lubomska, D., & Piechowicz, L. (2022). Into the Unknown: Microbial Communities in Caves, Their Role, and Potential Use. *Microorganisms*, *10*(2), Article 2. https://doi.org/10.3390/microorganisms10020222

Tomczyk-Żak, K., & Zielenkiewicz, U. (2016). Microbial Diversity in Caves. *Geomicrobiology Journal*, *33*(1), 20–38. https://doi.org/10.1080/01490451.2014.1003341

Wiseschart, A., Mhuanthong, W., Thongkam, P., Tangphatsornruang, S., Chantasingh, D., & Pootanakit, K. (2018). Bacterial Diversity and Phylogenetic Analysis of Type II Polyketide Synthase Gene from Manao-Pee Cave, Thailand. *Geomicrobiology Journal*, *35*, 518–527. https://doi.org/10.1080/01490451.2017.1411993

Zhou, J., Gu, Y., Zou, C., Mo, M. (2007). Phylogenetic diversity of bacteria in an earth-cave in Guizhou province, southwest of China. *Journal of Microbiology (Seoul, Korea)*, *45*(2). https://pubmed.ncbi.nlm.nih.gov/17483794/

Also, no insight is given about the possible contribution to the C cycling of fungi (see detailed comments).

We thank the reviewer for this input and will include a description of the fungal carbon cycle and its potential contributions to the cave carbon cycle. To better incorporate the role of arbuscular mycorrhiza fungi into our paper, we will add the below explanation to the discussion in section 5.1 (Line ~406).

The symbiotic relationship between arbuscular mycorrhizal fungi (AMF) and plant roots is ubiquitous globally, occurring in 80 % of terrestrial plants (Fitter et al., 2000). AMF grow on plant roots and form arbuscules within root cells, and hyphae which extend into the soil (Zhu & Miller, 2003). Plants benefit from this relationship because of the increased uptake of nutrients such as phosphorous and nitrogen and improved resistance to environmental stress through more stable soil structure and enhanced water absorption (Kiers et al., 2011; Boyno et al., 2024). The fungi benefit through access to plant derived carbohydrates for energy, and a stable environment for growth and reproduction (Kiers et al., 2011). The respiration of AMF can contribute to the subsurface sequestration of carbon (Treseder & Allen, 2008), and may indeed contribute to the subsurface $CO_2$ which we measured at Milandre cave. There appears to be a temperature dependance of roots with AMF colonization where respiration rates increase at higher temperatures (Atkin et al., 2008). We observe an increase in $CO_2$ concentration during the higher temperature summer months (Fig.3a, Fig.5a), which may be caused in part by a contribution of $CO_2$ from enhanced respiration of AMF. The isotopic composition of the $CO_2$ produced by the respiration of AMF is not currently well constrained. Studies analysing the $\delta^{13}C$ of individual AMF spores showed that the $\delta^{13}C$ broadly corresponded to the C3 or C4 plant that it was associated with (Nakano et al., 1999). Therefore, it can be assumed that $CO_2$ produced by AMF respiration is similar to that of the C3 host plant dominated catchment of Milandre cave. As carbon is fixed from modern atmospheric $CO_2$ during host plant photosynthesis, the $F^{14}C$ signature should be modern (~1). Hence, the respiration of AMF may contribute to the modern $CO_2$ measured in the boreholes in both meadow and forest sites (Fig.3 & 5).

Atkin, O. K., Sherlock, D., Fitter, A. H., Jarvis, S., Hughes, J. K., Campbell, C., Hurry, V., & Hodge, A. (2009). Temperature dependence of respiration in roots colonized by arbuscular mycorrhizal fungi. *New Phytologist*, *182*(1), 188–199. https://doi.org/10.1111/j.1469-8137.2008.02727.x

Boyno, G., Yerli, C., Çakmakci, T., Sahin, U., & Demir, S. (2024). The effect of arbuscular mycorrhizal fungi on carbon dioxide (CO2) emission from turfgrass soil under different irrigation

intervals. *Journal of Water and Climate Change*, *15*(2), 541–553. https://doi.org/10.2166/wcc.2024.482

Fitter, A. H., Heinemeyer, A., & Staddon, P. L. (2000). The impact of elevated CO2 and global climate change on arbuscular mycorrhizas: A mycocentric approach. *New Phytologist*, *147*(1), 179–187. https://doi.org/10.1046/j.1469-8137.2000.00680.x

Kiers, E. T., Duhamel, M., Beesetty, Y., Mensah, J. A., Franken, O., Verbruggen, E., Fellbaum, C. R., Kowalchuk, G. A., Hart, M. M., Bago, A., Palmer, T. M., West, S. A., Vandenkoornhuyse, P., Jansa, J., & Bücking, H. (2011). Reciprocal Rewards Stabilize Cooperation in the Mycorrhizal Symbiosis. *Science*, *333*(6044), 880–882. https://doi.org/10.1126/science.1208473

Nakano, A., Takahashi, K., & Kimura, M. (1999). The carbon origin of arbuscular mycorrhizal fungi estimated from δ13C values of individual spores. *Mycorrhiza*, *9*(1), 41–47. https://doi.org/10.1007/s005720050261

Treseder, K. K., & Allen, M. F. (2000). Mycorrhizal fungi have a potential role in soil carbon storage under elevated CO2 and nitrogen deposition. *New Phytologist*, *147*(1), 189–200. https://doi.org/10.1046/j.1469-8137.2000.00690.x

Zhu, Y.-G., & Miller, R. M. (2003). Carbon cycling by arbuscular mycorrhizal fungi in soil–plant systems. *Trends in Plant Science*, *8*(9), 407–409. https://doi.org/10.1016/S1360-1385(03)00184-5

Finally, it is difficult at times to read and understand the text. Perhaps there are also issues with figures (see specific comments). I have highlighted some of these difficulties arising from long sentences and graphics in the detailed comments. I believe the manuscript needs another "layer" of editing to improve readability.

We thank the reviewer for this feedback and their detailed comments below, which we will take on board. We are confident that, together with the feedback from reviewer 2 and our own final round of edits, we will be able to improve the readability of manuscript and figures.

I am not 100% sure about a general significance of the study, apart from the Grotte de Milandre context, because the study did not really discuss what the role of AM-vegetation in the soil and cave microbes (including fungi) could be. Yet, it is an honest study, with commendable efforts, which deserves to be published.

This study focuses on the chemistry of $CO_2$, and particularly the rare use of $^{14}C$ $CO_2$ analysis in the unsaturated zone to constrain carbon sources and turnover times. We agree that the role of fungi and other microbes play in cave carbon cycles is very interesting, and as pointed out by the reviewer, the topic of numerous existing studies. However, in our case, going into detail about the fungal and microbial diversity within Milandre cave was beyond the scope of this study. We aim to be able to distinguish between $CO_2$ of different ages, derived from organic and inorganic sources. We agree that expanding the discussion adding more details on processes in

the soil, unsaturated zone, and cave involving fungi, in-situ microbes, and soil lichens, will benefit the manuscript and we will make changes as described.

**Specific Comments**

line 75  Perhaps strike-slip fault is easier to understand for all.

We will change the sentence to use the term "strike-slip fault"

Also, same line, add: Late Jurassic (Middle-Oxfordian)  St-Ursanne Fm which overlies Early to Middle Oxfordian marls (Bischof et al., 2018).

We will change the sentence according to the suggestion.

Lines 80-8110 to 20 m deep or wide?

We will change the sentence to improve the clarity of the 20m deep unsaturated zone.

Fig. 1 Perhaps superimpose the name of the village on the aerial photo? It would also be useful to add the cross-section of the cave (see Vuilleumier et al., 2019 and 2021) within its geologic cross-section and plot the sampling point in the cross section. It would be easier to follow/understand the upstream-downstream datasets and the "cross-trip".  This would make it easier to interpret figures such as Fig. 8, especially if the sample sites number were reported both on the cross section and the figure(s).

We thank the reviewer for their suggestion. Location will be added to the aerial photo. However, it is hardly possible to represent the cross-section of the cave. Vuilleumier et al. focused only on the downstream part. At the scale of the entire cave, the cross section would basically be a horizontal black line. Therefore we prefer not to add this figure.

Line 114. ....at cool temperature...or in cool boxes ...in the refrigerator....

We will change this sentence for clarity "Prior to analysis, the samples were stored away from direct sunlight at cool indoor temperatures for a maximum of six weeks"

Line 125 I suggest to change into: ...and in single lines from a depth of 5 m (...).

We will change this sentence to that kindly suggested by the reviewer.

Line 170. I suggest to state somewhere here that F14C is the fraction modern (post-bomb peak 14C).

Thank you for this comment. We will add an explanation of the $F^{14}C$ notation to this section.

Fig. 2. I did not see the theoretical carbon reservoirs for the C4 (green). I see a grey bar.

We will change the colours of Figure 2 to be more legible.

Line 225 10'700 and all the following concentrations. I do not understand the , why 10'000 and let's say 6100?  Should be 10,700; 10,000; 6,100; 3,000...

We will change the ' to , in the numbers as suggested.

Line 355. I suggest to  change in "...shallow boreholes gas samples, which vary with depth, vegetation cover and seasons, suggest......."

Thank you, we will change this sentence to the suggestion.

Line 364.  I suggest: due to: i) slow, but persisting metabolic microbial activity at below freezing temperatures as low as....; ii) reduced, but persisting rhizosphere ...; reduced transportation....

 Thank you, we will change this sentence to the suggestion.

There should be a role for soil lichens and hyphae here as well. They seem to adapt to temperature changes (Lange, Green, 2005; Atkin et al., 2009). The role of lichen and fungi is still poorly explored in cave-based research. Yet, the ubiquitarian presence of fungi in the soil zone and, in particular, in the root zone, is probably important. Any idea if you have AM-plants (associated with Arbuscular mycorrhizal fungi)? AM are present in the majority of terrestrial

ecosystems on earth. For example, Glomus intraradices has been found in Swiss alpine meadows (Sykorova et al. 2007). Would you be able to comment on a potential role of fungi and soil lichens rather that only on that of  "microbes"?

Answered above.

Line 366-6  "Similar seasonal trends in CO2 concentrations have been reported in other studies (Billings et al; 1998; Pumpanen et al., 2003; Zhang et al., 2023). The CO2 concentrations in the Shallow 2 meadow boreholes do not show seasonality as  pronounced as... due to consistent CO2 accumulation year-round".

 Thank you, we will change this sentence to the suggestion.

Line 376 "with the meadow soils deeper, more... higher..." than?

Thank you, we will make the following change to improve the clarity of the sentence.

"The meadow and forested areas have developed contrasting soil compositions, with the meadow soils deeper, more compacted, and with a higher organic content than the forest soils, which are shallow, unconsolidated, and with more roots"

 Line 380-5.  Perhaps the work of Muhr et al., 2009 (Global Change Biology) on soil frost effects on soil respiration and its radiocarbon signature may shed some light here. Contrasting behaviour in moist and dry soils has also been observed for AM, which is an aspect that is not much investigated in karst science (see Lekberg, Y. and Koide, R.T., 2008. Effect of soil moisture and temperature during fallow on survival of contrasting isolates of arbuscular mycorrhizal fungi. Botany, 86(10), pp.1117-1124).

We thank the reviewer for this input and will include the potential influence of the difference in snow cover and soil frost on soil $CO_2$ in the meadow and forest sites. To better incorporate this into our paper, we will add the below explanation to the discussion section 5.1 (Line ~351).

Soil frost induced by colder temperatures can reduce heterotrophic respiration in soils. Experiments inducing deep frost in forested soils by removing snow cover caused a greater decrease in heterotrophic respiration during winter months compared to snow covered control soils (Muhr et al., 2009). Furthermore, frost-heave can physically damage the extraradical hyphae on plant roots, (Lekberg & Koide, 2008). Due to the shielding of the tree canopy, the forested areas in the Milandre catchment likely experience less snow cover compared to the open meadow where snow can accumulate freely. The forested soils may experience greater effects of frost because of the reduced snow cover during the winter, which may contribute to the decreased borehole $CO_2$ concentrations in winter in the forest boreholes (Shallow 1). As the

meadow soils likely experience reduced frost effects, the heterotrophic respiration is not impacted to the same extent and the forest soils and can accumulate $CO_2$ year-round, as seen in borehole Shallow 2.

Lines 397-401. Sentence is too long. One needs to breathe in-between.

We will split this sentence into two parts to improve readability.

Line 404. Upward movement of cave air....and downward ....

Thank you, we will change this sentence to the suggestion.

Line 450 dilution with atmospheric air

Thank you, we will change this sentence to the suggestion.

474- Dissolution of the host rock ...two extreme cases: completely closed and completely open systems. In a completely closed...

Thank you, we will change this sentence to the suggestion.

To be consistent, replace fully with completely.

Thank you, we will change this "fully" to "completely" in this sentence.

Remove solid, as a rock is, by definition...solid. Perhaps replace with carbonate ?

Thank you, we will change "rock" to "carbonate" in this sentence.

511-513. The sentence is a bit difficult to read. Perhaps stop at highest $F14C_{cave}$ . Then: When $F14C_{DIC}$ = 0.88 it results in an highly unlikely scenario on the basis...

Thank you, we will change this sentence to the suggestion.

520...though this hypothesis is highly unlikely (?).

Thank you, we will change this sentence to the suggestion.

529 (we considered or took a range...). There are several sentences mixing tenses. Please try to be consistent. Commonly, the past tense is used:...we took...we assumed...But perhaps this is not so important.

Thank you, we will edit the manuscript to be consistent with the tenses used.

I would omit "fresh".

Thank you, we will omit "fresh".

...fed through fracture flow. (fresh seems here to be associated with fracture...fresh fracture-flow?)

Thank you, we will be more specific and use "fracture flow".

595 One possibility: could gas trapped in formation water from the marls contribute CO2 (and methane)? (see Cailteau, 2008).

Cailteau (2008) measured the C-fluxes in boreholes which were likely out of equilibrium. We are not sure this analogy may hold in an open karst system but certainly worth to investigate in a future study.

607 "The topography likely reflects influences of the secondary porosity in the bedrock, which is possibly more highly fractured in the meadow doline formation compared to the forest leading to a greater effect of ventilation in the meadow area and an undisturbed modern 14C signature of forest soils. As the doline structure likely has a higher secondary porosity and is more highly fractured than the higher bedrock, ventilation effects may be more important here than at the other locations".

Thank you, we will change this section to the suggestion.

I did not understand the full meaning of the sentences. Does it mean that the formation of dolines is guided by fractures? But what is the "higher bedrock"? Also, given that the Formation(s) into which the cave develops is not completely pure limestone, the potential effects of the presence of marls should somewhat be considered.

We thank the reviewer for this comment and will improve the clarity of this sentence which tries to say that the forest is topographically higher than the meadow by a few meters.

We are not aware about any significant marly layer in the Milandre stratigraphy. The Upper Oxfordian marls with Astartes and Natices have been eroded on most of the catchment. The cave itself opens in the 60 to 80 m thick Middle Oxfordian St Ursanne Fm, which is a compact limestone with abundant corals.

After this, I gave up in the reviewing of sentences, because I feel it is not really the role of a reviewer.None

---

## Author Comment (AC2)

Response to Reviewer 2

Thank you for the supportive comments and suggestions to improve our manuscript. We have taken the suggestions onboard and will edit the manuscript according to the following responses.

This paper investigates the subsurface $CO_2$ dynamics in the Milandre cave karst system in northern Switzerland, focusing on seasonal and spatial variations. The study spans several years and involves analyzing $CO_2$ concentrations, stable carbon isotopic ratios ($\delta13C$), and radiocarbon ($14C$) compositions from various sources, including the outside atmosphere, boreholes, and cave air.

Key findings include i) that $CO_2$ concentrations in shallow boreholes are higher in summer due to increased respiration rates and lower in winter, ii) cave $CO_2$ concentrations and isotopic compositions are influenced by temperature-driven seasonal ventilation, iii) the cave air $CO_2$ levels were anti-correlated depending on where exactly in the cave the measurements were obtained, and iv) that the $CO_2$ in the cave is primarily sourced from modern soil respiration and aged organic material in the epikarst.

The research highlights the complexity of $CO_2$ dynamics in karst systems and the significant impact of seasonal and ecological factors on carbon cycling within the terrestrial Critical Zone. It is sound research reflecting the results of a long and intense monitoring campaign, and I recommend publication following the minor revisions suggested below.

There is a substantial amount of text devoted to describing the data, but a bit more describing the characteristics of the site with respect to carbon would be useful. For example, is there organic sediment within the cave site? How deep are the deep roots in the forest? Related to the above, I could not find the depth to the cave anywhere. It could well be here (apologies if so), but could it be included in the site description section? There is the sentence 'The unsaturated zone ranges between 40 and 80 m depth, with a saturated zone of ~ 20 m' but the cave could be developed anywhere within the unsaturated zone, so this does not let the reader know where the cave is relative to the surface. This is really critical for the interpretations presented here; for example, do the tree roots reach the cave? Also, why is the saturated zone only 20 meters? What is below the saturated zone?

Thank you for this detailed comment. We will improve the description of the Milandre site with more specific references to the cave sediments, unsaturated and saturated zone. The exact depth of the tree roots is not known, however there are no tree roots visible within the cave itself. We will be more specific with regards to this section.

The unsaturated reaches down to between 40 and 80 m below surface. The active karst level develops in the epiphreatic zone. The underneath phreatic zone extends for c. 20m to reach the low permeable Lower Oxfordian marls (Liesberg Mb) which act as a regional aquiclude. The site description will be amended to describe the stratigraphy more explicitly.

Line 48: Suggest rephrasing 'exclusively aged CO2 reservoirs' – a bit awkwardly phrased

Thank you, we will rephrase this to "...$CO_2$ reservoirs of a lower $F^{14}C$ corresponding to an older age" to improve the clarity.

Line 60 (and throughout): the comma in the 1,000s separator in the numbers should be on the bottom rather than the top.

Thank you, we will change the commas to be on the bottom when reporting numbers.

L69: 'provenance' not plural

Thank you, we will change this to "provenance".

L112: Elaborate – if it was built in house, why is there a brand name? Also spell out 'nondispersive infrared (NDIR)' for readers who may not be familiar with acronym

We thank the reviewer for this comment. The casing of the sensor was in-house built but the information is not really needed here.

We suggest adding this sentence "The $CO_2$ concentration was monitored during line flushing using a nondispersive infrared $CO_2$ sensor (SCD30, Sensirion, Switzerland) to ensure accurate sampling."

L138 and elsewhere: 'through trip' rather than 'cross trip'

Thank you, we will change "cross trip" to "through trip" throughout the manuscript for clarity.

L201-202: Technically this is all correct, but consider using either 'r' instead of the letter 'rho' or spell out 'rho' instead of using the letter. It looks a lot like the 'p' in p-value, and this would avoid some confusion. But up to the authors.

Thank you for catching this, we will replace all the "rho" symbols to "rho" the word.

L207: Use 'average' or 'mean' but be consistent. On this line both words are used in the same context. I'd recommend 'mean' when talking about a statistical quantity and 'average' when talking about something in general terms.

Thank you, we will change the use of "mean" and "average" as suggested throughout the manuscript.

Figure 2: Consider writing 'C3' and 'C4' and putting the text inside the filled boxes. Or connect the labels to the boxes with a connector line. The issue is that where they are now makes it seem like where the actual field is, and some readers might not notice the boxes. The same is very true for the blue atmosphere box – it is almost invisible. I definitely recommend drawing a line or an arrow between the text and the blue box in that case.

We agree that the legibility of this figure is lacking and will improve it by connecting the labels to the boxes and changing the colour of the boxes so it is more clear.

Figure 2: Use 'Upstream (Cave air)' and 'Downstream (Cave air)' in the legend, this might make it clear that these are in-cave measurements. It is clear from the text, but someone scanning the paper would be forgiven for thinking you had somehow taken these measurements from a stream.

Thank you for pointing this out! We will edit the legend accordingly.

Figure 3: It might be useful to show atmospheric CO2 concentration and chemistry here as well?

We agree that it would be useful to be able to compare to the atmospheric values, however adding this information to the figure would mean that the trends of the borehole $CO_2$ would be less clearly visible in the concentration (a) and $\delta^{13}C$ (b) graphs. We find it sufficient that the atmospheric values can be compared to the cave and borehole samples in Fig.2 and Fig.7.

Figure 3 and elsewhere: Why are there a maximum and minimum numbers for each timeslice? I assume that this is max and min of the monthly recorded values? I feel that this is probably

obvious - but i can't find it. So, if it is in the manuscript, it should be made more apparent, including here in the figure caption.

The maximum and minimum values are presented here because in these boreholes there are multiple sampling lines which sample over a depth range (see Table 1). However, due to the nature of the installation of these lines, we cannot distinguish between each depth. Therefore, we just present the maximum and minimum values which represent the range of $CO_2$ sampled from these boreholes. We were edit the text and the figure caption to more clearly state this.

L263: Is this truly referring to the epikarst (the relatively shallow zone of fractured rock directly beneath the soil), or is this bedrock within the unsaturated zone? Earlier, you use the term unsaturated zone, and the depths of the boreholes seem deeper than the epikarst, so I think you mean might mean unsaturated zone. The two have different meanings, and not interchangeable. Similarly, 'karst' refers to all the karstified limestone bedrock, regardless of whether it is saturated or unsaturated – it is not just the rock above the cave. Please change to appropriate terminology if these terms are used incorrectly. If they are used correctly, you might want to consider defining the term 'epikarst' since it is so often used incorrectly. This way readers will know for sure that what you are referring to as the epikarst is the irregular surface between the karst and the soil.

Thank you very much for pointing this out! Indeed, we use the term "epikarst" and unsaturated zone interchangeably which is not correct. We will edit the manuscript throughout to use the correct terms.

L418: How does borehole Deep 1 influence the correlation? I think needs to be reworded.

We thank the review for pointing this out. We think that this sentence is in fact an error, as there is no significant correlation between $pCO_2$ and MMT for Deep 1. We will delete this sentence accordingly.